META-RESEARCH

# Gender variations in citation distributions in medicine are very small and due to self-citation and journal prestige

**Abstract** A number of studies suggest that scientific papers with women in leading-author positions attract fewer citations than those with men in leading-author positions. We report the results of a matched case-control study of 1,269,542 papers in selected areas of medicine published between 2008 and 2014. We find that papers with female authors are, on average, cited between 6.5 and 12.6% less than papers with male authors. However, the standardized mean differences are very small, and the percentage overlaps between the distributions for male and female authors are extensive. Adjusting for self-citations, number of authors, international collaboration and journal prestige, we find near-identical per-paper citation impact for women and men in first and last author positions, with self-citations and journal prestige accounting for most of the small average differences. Our study demonstrates the importance of focusing greater attention on within-group variability and between-group overlap of distributions when interpreting and reporting results of gender-based comparisons of citation impact.
DOI: https://doi.org/10.7554/eLife.45374.001

**JENS PETER ANDERSEN\*, JESPER WIBORG SCHNEIDER, RESHMA JAGSI AND MATHIAS WULLUM NIELSEN**

## Introduction

Over the past four decades, the share of female graduates in medicine has increased from less than 10% to more than 50% in OECD countries, and recent statistics suggest near-parity in the representation of women and men as authors in medical research in Australia, Brazil, Chile, Europe and North America (*Huggett, 2017*; *OECD, 2019*). However, gender inequalities persist in the upper echelons of academic medicine. As of 2013, women constituted just 21% of full professors in the United States and just 23% in Europe, with the proportion of female department chairs and deans also remaining low (*European Commission, 2016*; *Lautenberger et al., 2014*).

These gender imbalances likely reflect myriad obstacles to women's career progress, including chilly and sometimes hostile work climates

(*Carr et al., 2003*; *Jenner et al., 2019*; *Pololi et al., 2013*), bias in recruitment and selection practices (*Van den Brink, 2011*), societal cultures that still expect a strongly gendered division of domestic labor (*Jolly et al., 2014*), an underrepresentation of women in last-author positions (*González-Álvarez and Cervera-Crespo, 2019*; *Jagsi et al., 2006*; *Lerchenmueller and Sorenson, 2018*), and disparities in research funding (*Jagsi et al., 2009*; *Sege et al., 2015*). Given that citation indicators are increasingly being used to inform tenure, hiring and funding decisions in many areas of the medical sciences, possible gender differences in citation impact have the potential to contribute to the perpetuation of these inequalities.

A survey of the literature revealed 22 papers on gender and citations in the medical sciences

**\*For correspondence:** jpa@ps.au.dk

published between 2006 and 2016 (see *Supplementary file 1*). The study designs, impact measures and statistics used in these papers are too heterogeneous for meta-analytical comparisons, and this literature is also characterized by notable variations in results depending on specialty, country, study design and type of citation indicator (h-index, citations per paper, cumulative citations, m-quotient and journal impact factor). Some studies report an average citation advantage in favor of men (see e.g. *Larivière et al., 2011*; *Nielsen, 2016*); others do not observe any notable gender difference (see e.g. *Mirnezami et al., 2016*; *Pagel and Hudetz, 2011*). Existing articles are in most cases based on convenience samples and limit their focus to single specialties or sub-specialties (16 out of 22), and the literature is characterized by a North American bias, with only five studies focusing on countries outside the US and Canada. Furthermore, only six of the papers report direct comparisons of the average number of citations per paper for male and female authors (*Housri et al., 2008*; *Larivière et al., 2011*; *Mirnezami et al., 2016*; *Nielsen, 2016*; *Pagel and Hudetz, 2015*; *Pagel and Hudetz, 2011*).

Researchers have also studied gender and citation distributions in fields other than medicine, and again these studies are characterized by ambiguous results that vary by geographical focus, time period and discipline. Some report average differences in favor of male authors (*Aksnes et al., 2011*; *Caplar et al., 2017*; *Eagly and Miller, 2016*; *Maliniak et al., 2013*), some report smaller average differences in favor of female authors (*Borrego et al., 2010*; *Long, 1992*; *van Arensbergen et al., 2012*), and some report no discernable gender difference (*Nielsen, 2017*; *Slyder et al., 2011*; *Symonds et al., 2006*).

A recurring limitation in the literature is the lack of attention paid to within-group variability and between-group overlap in citation distributions. Many studies rely on simple, mean-based comparisons to derive generic conclusions about gender differences (or similarities) in per-paper citation impact. This practice can be misleading for at least two reasons. First, the reported gender differences are generally small, and will, given the expected distribution for citation data, imply a great deal of overlap for women and men (we return to this below). Second, the lack

of attention to co-varying factors influencing a paper's likelihood of being cited (e.g. institutional affiliation, country-affiliation, self-citations and number of authors) will inevitably limit what can be learned from bivariate comparisons of this sort.

Here we report the results of a comprehensive, global analysis of possible gender variations in the per-paper citation impact of medical researchers. We analyzed 1,269,542 papers on disease-specific medical research published between 2008 and 2014. To reduce confounding and ensure balanced case-control groups, three matching covariates (institutional prestige, geographic location and medical specialty) were used to generate three datasets: sample 1 had female first authors as the case and male first authors as the control (*n*=1,018,665); sample 2 had female last authors as the case and male last authors as the control (*n*=653,233); and in sample 3, pairs of female first and last authors constituted the case group and all other author combinations were included in the control group (*n*=368,374). The outcome variable was field-normalized citations per paper, and regression analyzes were used to explore the influence of additional co-varying factors (self-citations, number of authors, international collaboration and journal prestige) on differences in per-paper citation impact (see Methods). Given the large sample size, global scope, and matched design, our study is less vulnerable to biases resulting from sample-specific variance, confounders and selection than previous studies in the medical field.

## Results

*Table 1* specifies the gender composition of the unmatched sample (*n*=1,269,542) by main specialty, institutional prestige and geographic location. Male researchers dominate all five main specialty groupings. Female last authors are underrepresented, in comparison to their representation in the global population, in all five groupings, but most notably in Surgical/Procedural specialties. The proportion of female first and last authors is highest in Latin America and lowest in South East Asia. Note here that numerous countries located in Eastern Asia have been excluded from the analysis due to unreliable gender disambiguation based on first-name and country information (see Methods for more details). In Western Europe and North America

**Table 1.** Women's share of authorships overall, across five main specialties, institutional prestige, and geocultural area.

| Overall | f_w | f_first | f_last | f_both |
|---|---|---|---|---|
| | 0.35 | 0.40 | 0.26 | 0.15 |
| **Main specialty** | f_w | f_first | f_last | f_both |
| Basic science | 0.39 | 0.46 | 0.30 | 0.18 |
| Hospital based | 0.37 | 0.43 | 0.28 | 0.16 |
| Medical | 0.33 | 0.38 | 0.24 | 0.13 |
| Pediatric | 0.46 | 0.52 | 0.37 | 0.24 |
| Surgical/procedural | 0.29 | 0.32 | 0.21 | 0.11 |
| **Institutional prestige** | f_w | f_first | f_last | f_both |
| Top-100 University | 0.36 | 0.42 | 0.27 | 0.16 |
| Other university | 0.35 | 0.39 | 0.25 | 0.14 |
| **Geographic location** | f_w | f_first | f_last | f_both |
| Arab countries | 0.33 | 0.34 | 0.27 | 0.16 |
| Commonwealth of Independent States | 0.40 | 0.45 | 0.30 | 0.17 |
| East Asia | 0.19 | 0.19 | 0.09 | 0.04 |
| Latin America | 0.46 | 0.52 | 0.39 | 0.25 |
| North America | 0.36 | 0.40 | 0.27 | 0.15 |
| Oceania | 0.40 | 0.48 | 0.31 | 0.20 |
| South and Central Europe | 0.40 | 0.44 | 0.31 | 0.18 |
| Sub-Saharan Africa | 0.36 | 0.39 | 0.31 | 0.20 |
| South-West Asia | 0.29 | 0.31 | 0.24 | 0.10 |
| Western Europe | 0.35 | 0.42 | 0.24 | 0.14 |

f_w is the weighted proportion of women per paper, f_first the proportion of female first authorships, f_last the proportion of female last authorships, f_both the proportion of papers where women are both first and last authors.
DOI: https://doi.org/10.7554/eLife.45374.002

the proportions of female first and last authors lie close to the Global averages.

Citation impact per paper is measured by field-normalized citation scores with a four-year fixed citation window (NCS). Using NCS as the outcome variable strengthens our matched design by adjusting for sub-specialty variations in citation practices. NCS is a continuous outcome variable. It is calculated by dividing the citations accrued by a paper within the first four years after publication with the expected citation score of other papers published in the same year and field (*Waltman et al., 2012*). Fields are delineated using the same article-level classification system as the Leiden Ranking. This classification system consists of 4,047 micro-level clusters of publications and offers one of the best, current approaches to item-oriented field normalization (*Waltman and van Eck, 2012*). This item-oriented field normalization procedure allows for comparison of papers published in different sub-fields, with different publication dates.

*Figure 1* displays the density distributions of log-transformed citation scores for the matched sets of papers with female first authors and male first authors (Sample 1), female last authors and male last authors (Sample 2), and female first and last authors and other gender combinations of first and last authors (Sample 3). For all distributions, the absolute uncertainty of the mean is between 0.001 and 0.005. On average, papers with female first authors are cited 8.7% less than papers with male first authors (Sample 1. Female first authors: $n = 509,330$; $\bar{x} = 1.16$; $\sigma = 1.83$; $\tilde{x} = 0.73$. Male first authors: $n = 509.335$; $\bar{x} = 1.27$; $\sigma = 2.00$; $\tilde{x} = 0.76$); however, the overlap between the two distributions is extensive (Cohen's $d = -.06$; Weitzman's $\Delta = 95.4\%$; *Weitzman, 1970*). Papers with female last authors are cited 6.5% less than papers with male last authors (Sample 2. Female last authors: $n = 326,611$; $\bar{x} = 1.16$; $\sigma = 1.93$; $\tilde{x} = 0.72$. Male first authors: $n = 326,622$; $\bar{x} = 1.24$; $\sigma = 1.93$; $\tilde{x} = 0.76$); again, the overlap between the two distributions is extensive (Cohen's $d = -.04$;

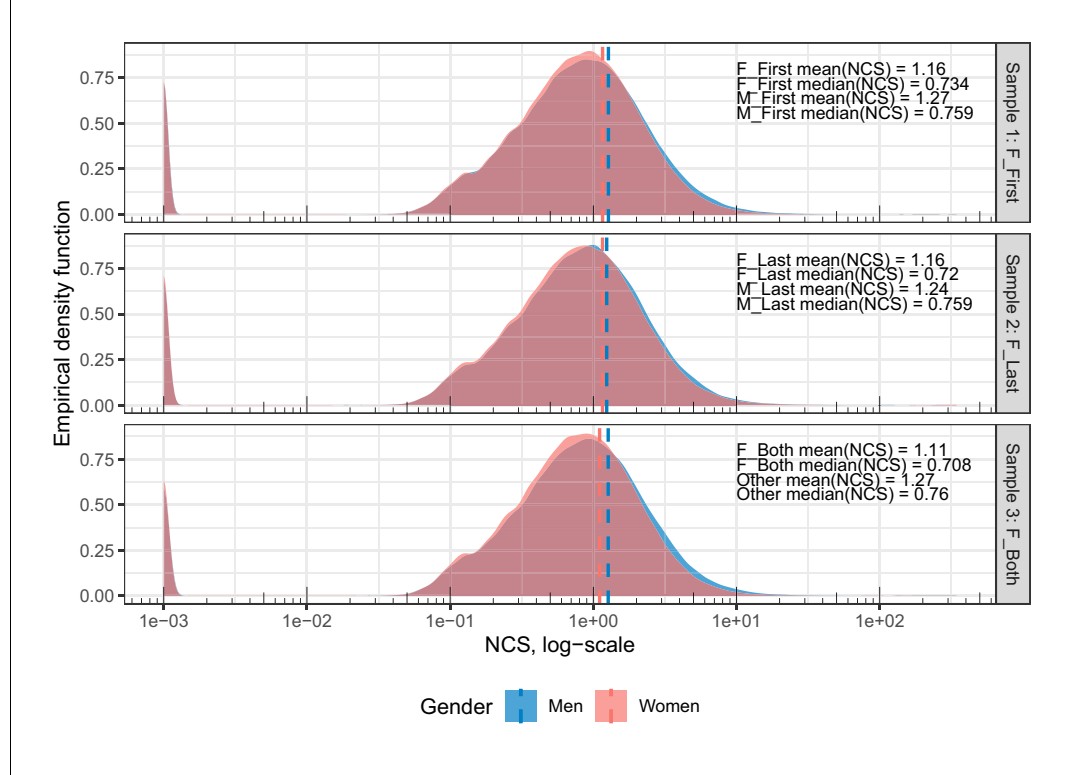

**Figure 1.** Density distributions of the log-transformed, per-paper NCS for the matched set of male and female first authors (Sample 1), female and male last authors (Sample 2), and female first and last authors vs. other author combinations (Sample 3). Dashed lines indicate the mean NCS for each sample. The y-axis indicates the proportion of papers found in that area of the NCS, equivalent to a smoothed histogram. The x-axis gives the per-paper NCS on a log-transformed scale. For all distributions, between-group overlap is extensive (93.1% to 95.6%). The difference between men and women is most clearly seen in the exceptionally highly cited studies, of which there are relatively few. Please note that. 001 (=1e-03) has been added to NCS in order to include uncited papers. The left-most peak in each sample represents uncited papers. The proportion of uncited papers per sample is 5.7%, 6.1%, and 5.9% for the case papers and 5.9%, 5.9%, and 5.8% for the control papers.

DOI: https://doi.org/10.7554/eLife.45374.003

Weitzman's $\Delta$ = 95.6%). Finally, papers in which both the first and last authors are female are cited 12.6% less than papers with other gender combinations (Sample 3. Female first and last authors: $n$ = 184,183; $\bar{x}$ = 1.11; $\sigma$ = 1.75; $\tilde{x}$ =0.71. Other combinations: $n$ = 184,191; $\bar{x}$ = 1.27; $\sigma$= 2.28; $\tilde{x}$ =0.76); again, the overlap between the two distributions is extensive (Cohen's $d$ = −.08; Weitzman's $\Delta$ = 93.1%).

To obtain a closer approximation of the observed gender variation on the right side of the distribution curves (*Figure 1*), we calculated the percentage share of case papers among the top 5% and top 10% most cited papers in each sample. Note here that the case and control-papers, given our matching approach, each comprise 50% of all papers in Samples 1, 2, and 3. Our calculations showed that papers with female first authors comprise 43.6% of the top

5% most cited and 45.5% of the top 10% most cited in Sample 1. Papers with female last authors comprise 45.8% of the top 5% most cited and 46.8% of the top 10% most cited in Sample 2. Finally, papers in which both the first and last authors are female comprise 41.1% of the top 5% most cited and 43.5% of the top 10% most cited in Sample 3.

While these results indicate large within-group variation and very small between-group differences, they are consistent with previous reports of a slight average citation advantage for papers by male lead authors. We decided to use regression analyses to explore the underlying variations that may drive this average difference.

We ran Tweedie regressions to estimate the residual effect of author gender after adjusting for self-citations, numbers of authors per paper,

field-normalized journal impact (MNCS journal) and international collaboration. MNCS journal is calculated as the mean-normalized citation score (NCS) of all papers published in a journal in a given year; in this case the same year as the observed paper. MNCS journal is essentially a measure of journal prestige that takes into account that most journals publish papers in more than one field, and that different fields have different citation characteristics. (See Methods for a discussion of the relationship between NCS and MNCS).

Exponentiated beta coefficients and 95% confidence intervals for the three models with NCS as outcome are displayed in *Figure 2*. The exponentiated beta coefficients should be interpreted as, *ceteris paribus*, the predicted relative change in the outcome resulting from a one unit increase in the predictor (*Coates et al., 2018*). For instance, an exponentiated coefficient of 0.95 for the case variable in Sample 1 (male first author = 0, female first author = 1) would indicate that female first authors, on average, receive 0.95 times the citations accrued by comparable papers with male first authors, hence on average 5% fewer citations.

The numeric input variables have been rescaled by dividing by two standard deviations (*Gelman, 2008*). We did this to allow the numeric inputs (i.e. MNCS journal, N authors, self-citations) to be interpreted on the same scale as the binary case variables (i.e. F_first, F_last, F_both). The standardized coefficients should be interpreted as two-standard deviation changes on a logit scale, from a low value to a high value (*Gelman, 2008*). *Figure 2—source data 1* summarizes the exponentiated values for both the direct and standardized coefficients.

The models indicate very small residual effects of author gender on citation impact per paper. The exponentiated coefficient for the case variable, F_first in Sample 1 (F_first: female first author=1, male first authors=0) is 0.98 (CI=0.98–0.98). The exponentiated coefficient for the case variable in Sample 2 (F_last: female last author=1, male last author=0) is 0.99 (CI=0.98–0.99), and the exponentiated coefficient for the case variable in Sample 3 (F_last: female first and last authors=1, all other author constellations=0) is 0.96 (CI=0.96–0.97). MNCS journal and self-citations are the strongest predictors in the models although their effect sizes can be considered small. The two remaining covariates (N authors and International collaboration) both have exponentiated coefficients close to one, indicating small effects.

*Figure 3* displays the estimated marginal means (EM-means) and 95% CIs for the case variables in Samples 1, 2 and 3. The EM-means are used to report the predicted, average citation score per paper for each group, adjusting for self-citations, number of authors, MNCS journal and international collaboration. In accordance with *Figure 1*, we observe only very small differences between the EM-means for the case and control groups in the three samples (*Figure 3*). Robustness checks were carried out to examine the sensitivity of the regression results to alternative model and sample specifications (for specifications see Methods). All of these analyses yielded qualitatively similar results. All of the models indicated very small residual effects of author gender on citation impact per paper (see *Figure 2—source datas 2–4*).

To examine which of the covariates that vary the most by lead-author gender, we ran three logistic regressions with F_first as outcome in Sample 1, F_last as outcome in Sample 2, and F_both as outcome in Sample 3. The results are presented in *Figure 4*. Again, the regression inputs have been rescaled by dividing by two standard deviations to allow for meaningful comparisons of the binary and numeric variables. *Figure 4—source data 1* summarizes both the direct and the standardized coefficients. Self-citations have the largest standardized coefficients in all three samples, followed by N authors in Samples 1 and 2, and MNCS journal in Sample 3. We observe no discernable effect of international collaboration on the outcome variable in any of the samples. The results indicate that papers with high self-citation rates and high MNCS journal scores are less likely to be led by female authors than male authors in all samples. Notice that the effect sizes are small. Further, papers with a high value of N authors are slightly more likely to have a female first author than a male first author, and slightly less likely to have a female last author than a male last author. As indicated in *Figure 2*, N authors has exponentiated coefficients extremely close to 1.0 in the Tweedie regressions with NCS as outcome. Hence, we restrict our focus to self-citations and MNCS journal in the remaining part of the analysis.

Descriptive analysis indicates larger average gender differences in self-citation rates compared to MNCS journal scores in Samples 1, 2 and 3 (*Table 2*), but the standardized mean differences for both variables are very small, and the percentage overlaps are extensive.

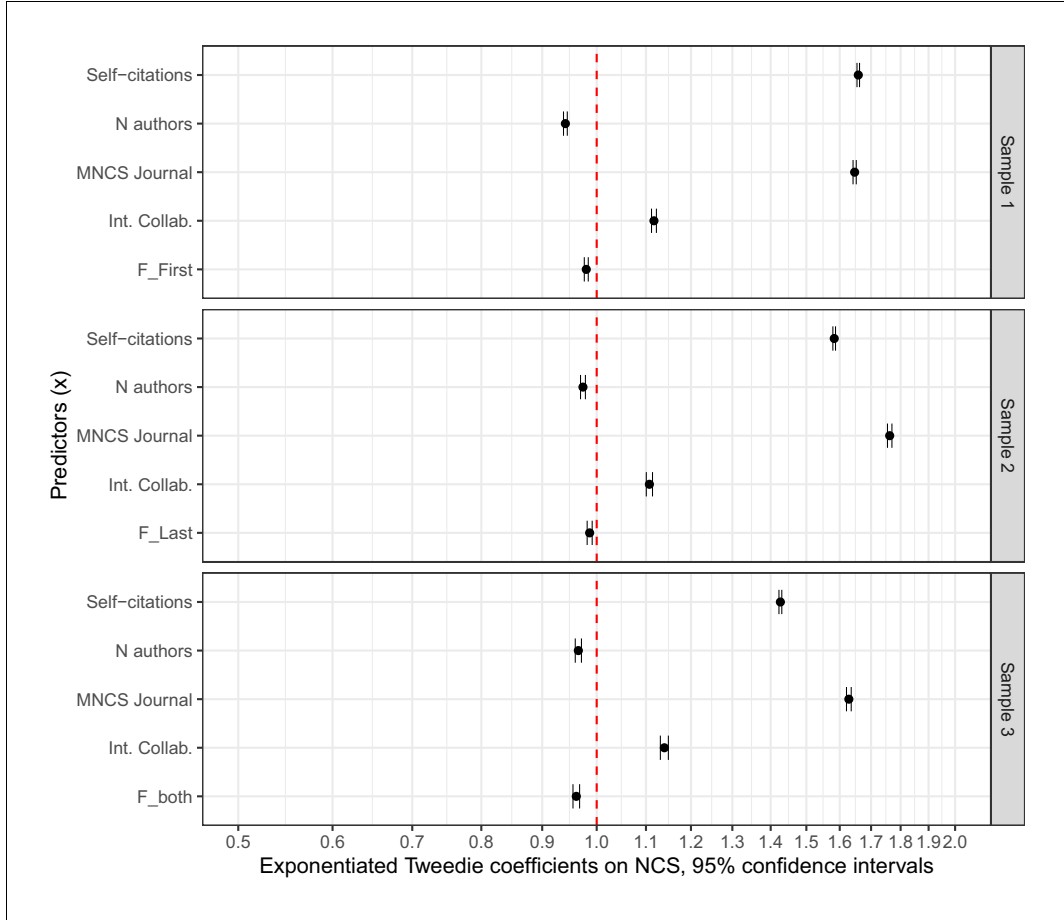

**Figure 2.** Standardized, exponentiated coefficients for the predictors included in the Tweedie regressions. Error bars represent 95% confidence intervals (see *Figure 2—source data 1* for estimate specifications and dispersion parameters). All regressions are based on matched samples. Sample 1 compares papers with female first authors to those with male first authors. Sample 2 compares papers with female last authors to those with male last authors. Sample 3 compares papers with female first and last authors to those with other author combinations. Values are on a logarithmic scale. The figure indicates very small residual effects of gender on NCS (case variables: F_First, F_Last and F_Both).

DOI: https://doi.org/10.7554/eLife.45374.004

The following source data is available for figure 2:

**Source data 1.** Tweedie regression results.
DOI: https://doi.org/10.7554/eLife.45374.005

**Source data 2.** Regression results for Tweedie regressions on the full, unmatched sample, using NCS as outcome.
DOI: https://doi.org/10.7554/eLife.45374.006

**Source data 3.** Regression results for the three negative binomial regressions with times cited (CS) as outcome.
DOI: https://doi.org/10.7554/eLife.45374.007

**Source data 4.** Tweedie regression of standardized parameters, using MNCS Journal quantiles rather than measurements.
DOI: https://doi.org/10.7554/eLife.45374.008

To obtain a closer approximation of the extent to which self-citations may contribute to explain the observed gender variation on the right side of the distribution curves in *Figure 1*, we plotted the average proportion of per-paper self-citations (number of per-paper self-citations/raw per-paper citation scores) in 5% intervals from the quantile of least cited papers to the quantile of top cited papers in Samples 1, 2 and 3. As displayed in the upper panel of *Figure 5*, the average proportion of self-citations for papers in the top 5% bin is ~15% in all three samples. This implies that at least part of the gender variation observed on the right side of

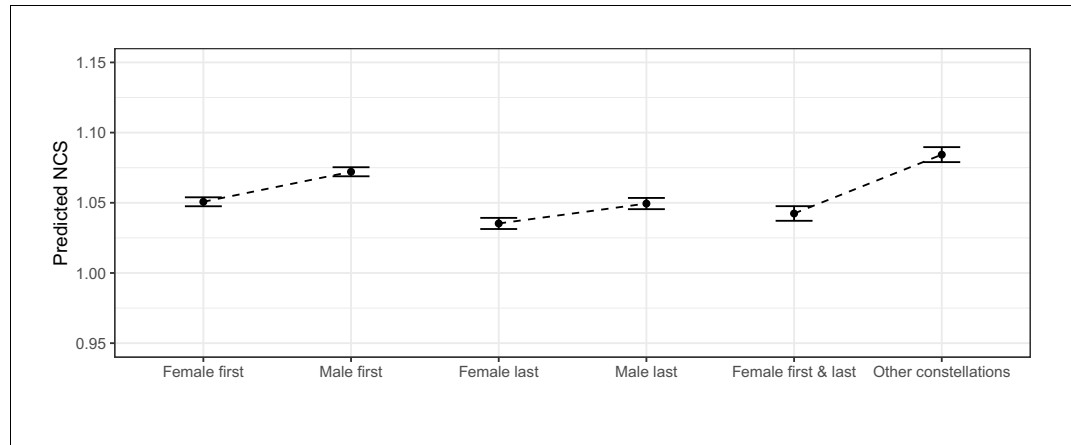

**Figure 3.** Plot of estimated marginal means for the case and control groups in Samples 1, 2 and 3. The error bars display 95% confidence intervals. The figure visualizes the predicted, average, differences in per-paper citation scores for the case and control groups after adjusting for self-citations, number of authors, MNCS journal, and international collaboration. Sample 1 compares papers with female first authors to those with male first authors. Sample 2 compares papers with female last authors to those with male last authors. Sample 3 compares papers with female first and last authors to those with other author combinations. Note that the y-axis has a restricted span from. 95 to 1.15. The comparisons indicate trivial, average gender differences.

DOI: https://doi.org/10.7554/eLife.45374.009

the curves in *Figure 1* may be attributable to average gender differences in self-citation rates per paper. It should also be noted that our citation indicators are calculated with a four-year window, which may contribute to explain the relatively large proportion of self-citations in the samples.

Finally, to examine the observed gender variation in MNCS journal scores in closer detail, we plotted the average proportion of case papers (i.e. papers with female first authors in Sample 1, papers with female last authors in Sample 2, and papers with female first and last authors in Sample 3) in 5% intervals – from the papers with the lowest to the highest MNCS journal scores. Note again that the case papers and the control papers each comprise 50% of all papers in Samples 1, 2, and 3. As displayed in the upper panel in *Figure 6*, the relative proportion of case papers is slightly overrepresented on the left side and in the middle part of the x-axis and slightly underrepresented on the right side. The relative proportion of case papers drops below 50% at the 80th percentile. The most notable drop occurs at the 90th–95th percentile. This indicates that most of the observed, average gender differences in MNCS journal scores are due to an underrepresentation of female-led papers in the most high impact publication outlets. Of the papers published in the top 5% outlets with the highest MNCS journal

scores, papers with female first authors and female last authors comprise 45% and 44%, respectively, while papers with female first and last authors comprise 41%.

## Discussion

Decision-makers in academic medicine increasingly use citation-based metrics to evaluate scholarly merits and allocate individual opportunities and rewards. Examining data to discern systematic gender differences in medical researchers' citation impact is therefore more important than ever. However, existing evidence from the medical sciences falls short of providing tangible guidance for policy on this issue. In this study, we carried out a controlled, large-scale, global gender comparison of lead authors' average citation impact per paper in disease-specific medical research.

Descriptive analysis indicated average gender differences of 6.5 to 12.6% in the three matched samples. However, the standardized mean differences were very small, and the percentage overlaps between male and female distributions were extensive.

In regressions adjusted for international collaboration, self-citations, number of authors, and field-normalized journal impact, the average citation impact per paper was close to identical for women and men, irrespective of first and last author combinations. Additional analyses

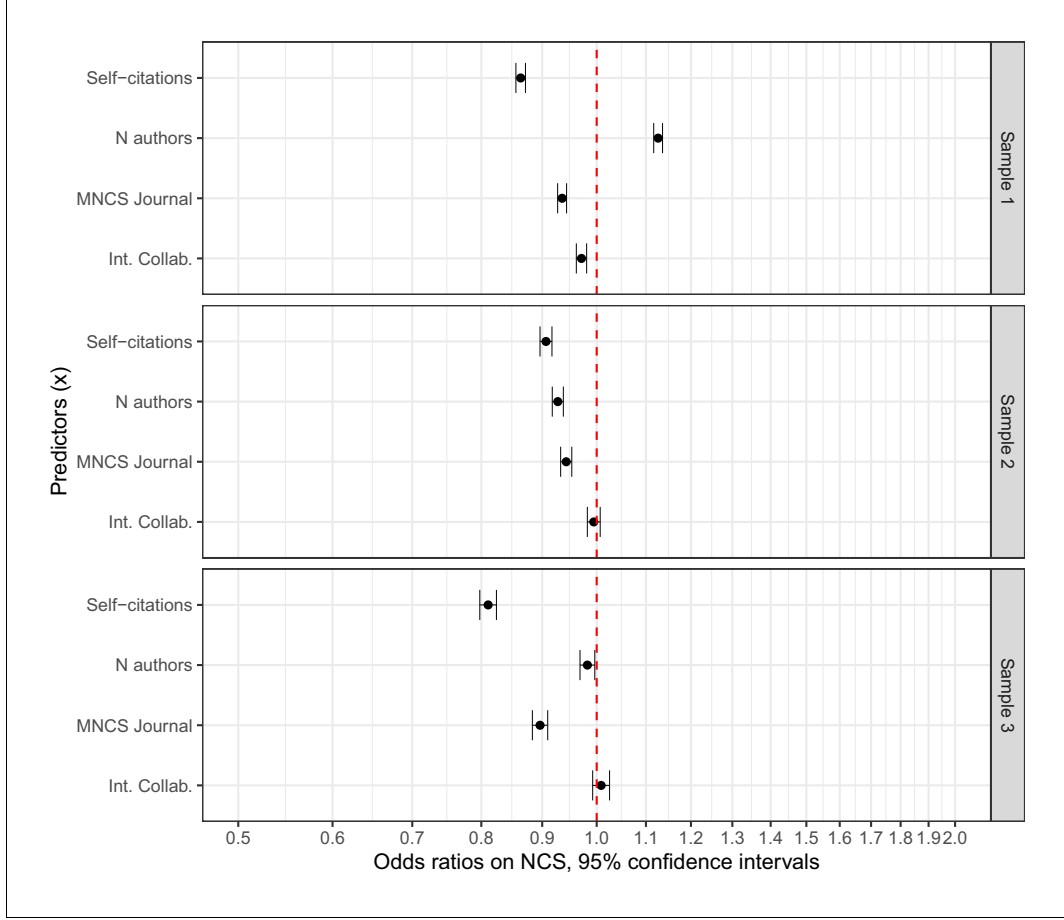

**Figure 4.** Odds ratios for the standardized predictors included in the logistic regressions. Error bars represent 95% confidence intervals (see *Figure 4—source data 1* for information on estimates and dispersion parameters). All regressions are based on matched samples. Sample 1 compares papers with female first authors to those with male first authors. Sample 2 compares papers with female last authors to those with male last authors. Sample 3 compares papers with female first and last authors to those with other author combinations. The figure indicates that self-citations is the variable that varies the most along gender lines in all three samples, albeit the effects can be considered small.

DOI: https://doi.org/10.7554/eLife.45374.010

The following source data is available for figure 4:

**Source data 1.** Logistic regression results.

DOI: https://doi.org/10.7554/eLife.45374.011

indicated that self-citations and field-normalized journal impact accounted for most of the average differences observed in bivariate gender comparisons.

Compared to previous studies, these findings align with what is known about "decline effects" (*Ioannidis, 2005*). As sample sizes become larger and study designs more advanced, effect sizes tend to decline towards minuscule effects.

Here, it is relevant to highlight that the gender effect attributable to self-citations likely reflects a generation effect, with more senior male authors having more publications to self-

cite. For instance a recent analysis of 1.5 million life-science papers demonstrates that observed gender differences in self-citation rates are leveled out when adjusting for each author's prior publication output (*Mishra et al., 2018*; see also *King et al., 2017*).

Given this latent generation effect, one could argue that it is somewhat surprising that we end up observing very small residual effects of author gender on per-paper citation impact. If the male authors in our sample, on average, are older and more established researchers, one

**Table 2.** Means, standard deviations, medians, Cohen's *d*, and Weitzman's *Δ* for case-control comparisons of self-citations and MNCS journal in Samples 1, 2 and 3.

| | | $\bar{X}$ **case (σ)** | $\bar{X}$ **control (σ)** | $\tilde{X}$ **case** | $\tilde{X}$ **control** | **d** | **Δ** |
|---|---|---|---|---|---|---|---|
| Sample 1 | Self-citations | 1.91 (3.18) | 2.16 (3.93) | 1 | 1 | -0.07 | |
| | MNCS journal | 1.16 (.90) | 1.21 (1.04) | .99 | 1.00 | -0.05 | 96.4% |
| Sample 2 | Self-citations | 1.84 (3.22) | 2.08 (3.77) | 1 | 1 | -0.07 | |
| | MNCS journal | 1.14 (.98) | 1.20 (.99) | .98 | 1.00 | -0.06 | 95.6% |
| Sample 3 | Self-citations | 1.74 (2.84) | 2.13 (3.91) | 1 | 1 | -0.11 | |
| | MNCS journal | 1.12 (.97) | 1.20 (1.02) | .97 | 1.0 | -0.08 | 93.4% |

Cohen's *d* and Weitzman's *Δ* are calculated with two and one decimal respectively. Weitzman's *Δ* is not calculated for self-citations, as it is a discrete count variables. For sample 1, female first authors is the case and male first authors is the control. For Sample 2, female last authors is the case and male last authors is the control. For Sample 3, female first and last authors is the case and other combinations are the control.

DOI: https://doi.org/10.7554/eLife.45374.012

would expect them to have a slight citation advantage in a cross-sectional analysis like ours.

Note also that women may still receive less credit for their citations due to gender-based double standards in evaluative judgments (*Botelho and Abraham, 2017*; *Moss-Racusin et al., 2012*). For instance, a sophisticated analysis of scientists receiving early-career grants from the National Institutes of Health in the US (1985–2009) shows that women compared to men gain fewer returns on citations in terms of transition time from a postdoc grant to an R01 grant. Adjusting for a large number of factors, women on average spent one year longer transitioning to an R01 grant than men with the same number of citations (*Lerchenmueller and Sorenson, 2018*).

The observed differences in the relative proportion of female- and male-led papers published in the journals with the highest field-normalized impact echo the findings of previous research in the medical sciences (*Jagsi et al., 2006*; *Lerchenmueller and Sorenson, 2018*). Part of this difference may be due to the generation effect described above. Other possible explanations may be that female lead authors are less likely to submit their research to journals with high impact factors (*Berg, 2017*; *Nature Neuroscience, 2018*), or that they have slightly lower success rates in peer review, e.g. due to gender bias or topic bias in journals with high impact factors. In the future, we hope to see closer examinations of the mechanisms driving this gender gap. This issue is critical given the strong emphasis on publishing in journals with high impact factors in evaluations of the performance of individual researchers (*McKiernan et al., 2019*).

Building diverse and inclusive research organizations requires careful attention to the mechanisms perpetuating gender inequalities in the higher academic ranks. The results reported here demonstrate that gender differences in per-paper citation impact are a negligible factor in this stratification process. Instead, universities and research leaders should develop strategies for effectively counteracting the broader societal and institutional barriers to the scientific advancement of women, including chilly and hostile work climates (*Carr et al., 2003*; *Pololi et al., 2013*), gender bias in recruitment and selection practices (*Van den Brink, 2011*) and work-family conflicts in the early stages of the academic career (*Jolly et al., 2014*).

Counteracting such barriers may be critical to improving the rigor and precision of medical research. For instance, an association between the representation of women as last authors (in combination with male first authors) and adequate statistical power in clinical trials was reported recently (*Otte et al., 2018*). Another recent study suggests a robust positive correlation between the attention to gender- and sex-related variations in medical research and the involvement of women as first and last authors (*Nielsen et al., 2017*).

Our study has some limitations. First, the analysis excluded articles with first or last authors from 18 countries, due to unreliable gender determination (see Methods). Despite a relatively small reduction in sample size (7.2% of the total population), this exclusion implies that countries located in East Asia and Sub-Saharan Africa, are underrepresented in the analysis, and the results should be interpreted accordingly.

Second, this study limited its focus to average gender differences in citation impact per paper.

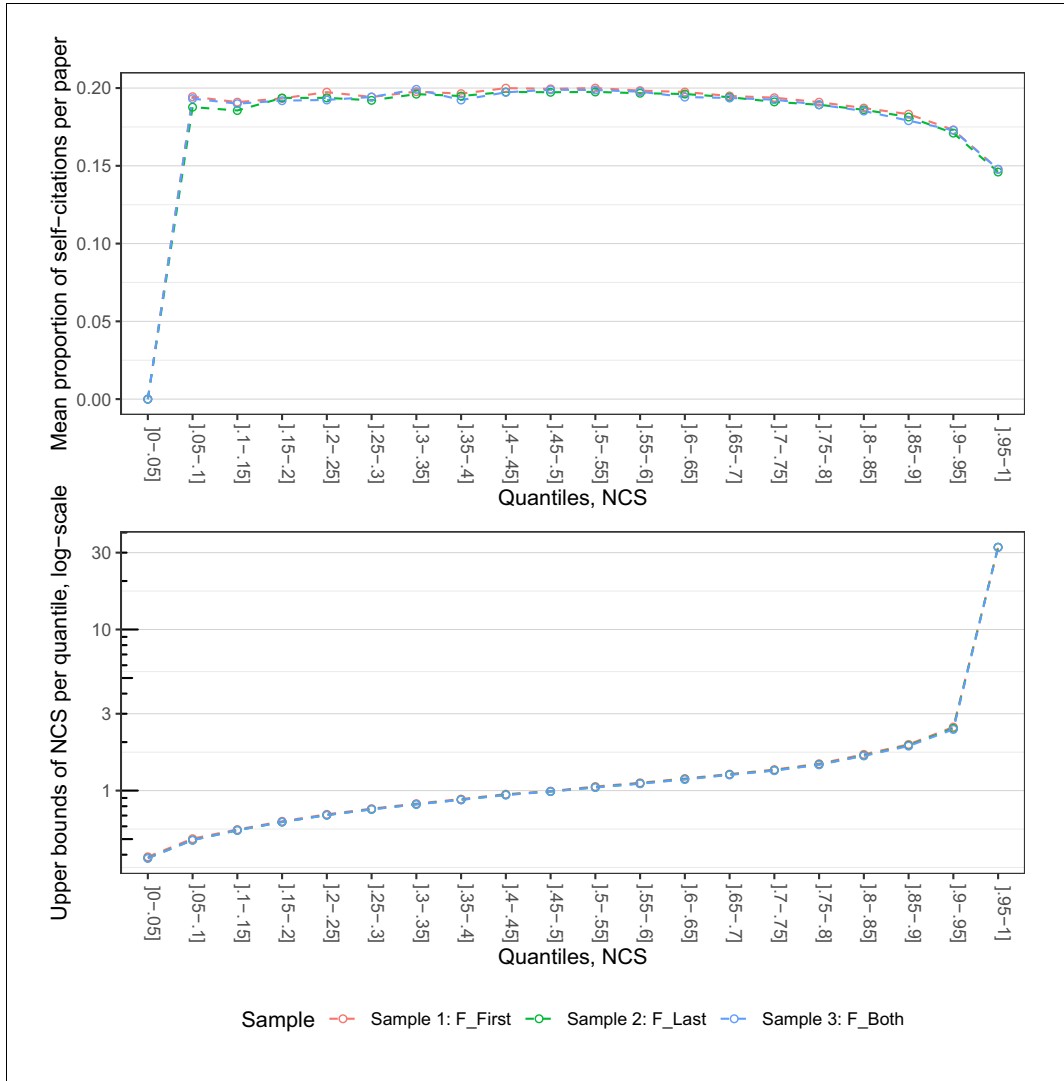

**Figure 5.** The upper panel shows the distribution of self-citations by five-percentile bins of NCS for each sample. The average proportions of self-citations are given on the y-axis, the five-percentile bins of NCS on the x-axis. The lower panel displays the distribution of the upper bounds of NCS across the five-percentile bins of NCS. The upper bounds of NCS are given on the y-axis, and the five-percentile bins on the y-axis.
DOI: https://doi.org/10.7554/eLife.45374.013

This focus precludes us from drawing any conclusions on potential long-term differences in male and female researchers' cumulative citation impact. If men, for instance, have higher average publication rates (e.g. due to shorter career breaks, more collaborative research articles, more funding and more people in their labs), this would imply that their average cumulative citation impact would be higher as well. Hence, while differences in per-paper citations appear to be a negligible factor in the perpetuation of gender inequalities in academic medicine, disparities in publication rates may still play a role – especially at the early-career stages and

especially in evaluation systems where publication rates have a strong influence on decisions about tenure, hiring and funding. Future research could use author-disambiguation algorithms to compare the cumulative citation impact and publication rates of large samples of individual researchers over time, adjusting for institutional affiliation, country affiliation, research area, scientific age and other relevant factors.

Third, per-paper citation impact represents a very specific proxy of scholarly impact. In practice, research evaluators at funding organizations and universities may use citation indicators

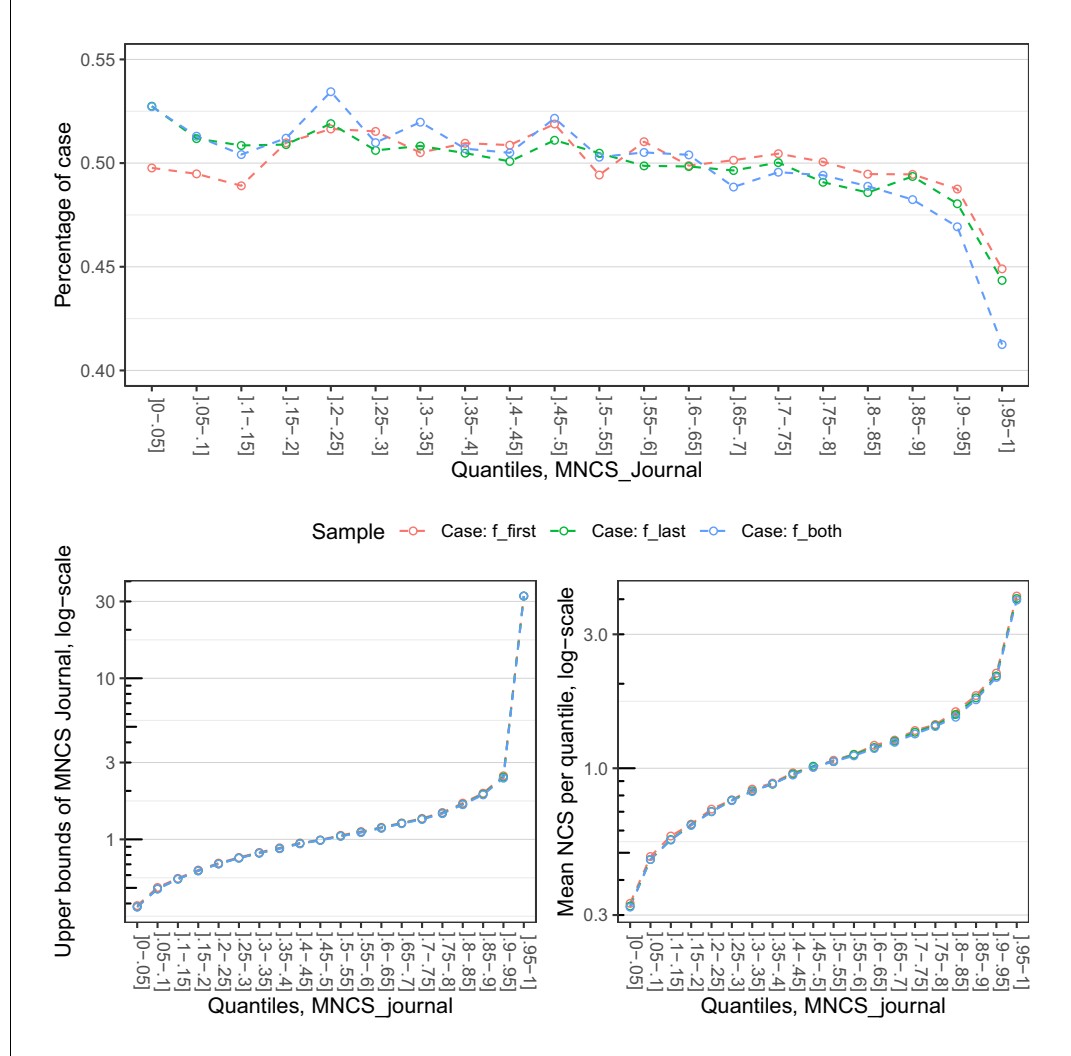

**Figure 6.** The upper panel shows the proportions of papers with female first authors in Sample 1, female last authors in Sample 2, combinations of female first and last authors in Sample 3, by five-percentile bins of MNCS. The proportions of case papers are given on the y-axis, and the five-percentile bins of MNCS journal on the y-axis. The lower-left panel displays the upper bounds of MNCS journal by five-percentile bins of MNCS journal for each sample, while the lower-right panel shows the mean NCS by five-percentile bins of MNCS journal for each sample. The upper bounds of MNCS journal (left) and Mean NCS (right) are given on the y-axes, and the five-percentile bins of MNCS journal on the x-axes.

DOI: https://doi.org/10.7554/eLife.45374.014

in other ways, e.g. by restricting their focus to the five most influential publications of an applicant. Indeed, having a few highly cited papers may in some evaluative contexts do more for a researcher's career progression than having a higher than average per-paper citation impact.

Fourth, the relatively small gender differences observed in the descriptive analysis should be interpreted with caution. Such results are sensitive to generic noise in the data (*Schneider, 2013*), and inherent uncertainties associated with statistical inferences based on non-random samples of "found data" (*Freedman et al., 2003*).

In conclusion, our results demonstrate that adjusting for co-varying factors, men and women in first and last author positions are cited at similar rates. The analysis presented here raises concerns that at least parts of the gender differences reported in prior research may be distorted by methodological limitations and imprecision in how the results are interpreted. We acknowledge the critical importance of recognizing even the small drawbacks that can add

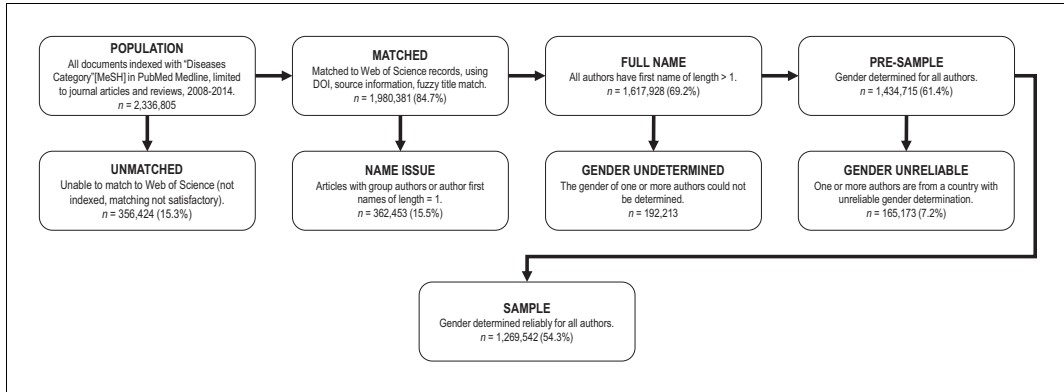

**Figure 7.** Flowchart of data collection, inclusion and exclusion.
DOI: https://doi.org/10.7554/eLife.45374.015

The following source data and figure supplements are available for figure 7:

**Source data 1.** Excluded countries due to unreliable gender assignments from first name.
DOI: https://doi.org/10.7554/eLife.45374.019
**Source data 2.** List of specialty and main specialty designation, and number of papers per specialty for the full sample.
DOI: https://doi.org/10.7554/eLife.45374.020
**Source data 3.** Groupings of countries by geographical region.
DOI: https://doi.org/10.7554/eLife.45374.021
**Figure supplement 1.** Percentage of papers per journal included in the analysis.
DOI: https://doi.org/10.7554/eLife.45374.016
**Figure supplement 2.** Reliability of gender assignment per country, shown as the rank of countries.
DOI: https://doi.org/10.7554/eLife.45374.017
**Figure supplement 3.** Proportion of papers with gender assignment for all authors.
DOI: https://doi.org/10.7554/eLife.45374.018

up over time and become cumulative disadvantages for women in science (*Caplan, 1993*; *Cole and Singer, 1991*). However, our study demonstrates the importance of focusing greater attention on within-group variability and between-group overlap of distributions when interpreting and reporting results.

## Methods

*Figure 7* displays the data-selection process. Peer-reviewed articles published between 2008 and 2014 were collected in PubMed Medline. To target core medical research and enable exact matching based on primary medical specialty, we needed information on the disease-specific Medical Subject Headings assigned to each paper. Hence, the initial sample was limited to records indexed with the broad MeSH descriptor "Diseases Category" (*n*=2,336,805). Eligible PubMed records were matched to article metadata in Web of Science (WoS) (citation data, author first names and affiliations), using Publication identifiers (PMID, DOI) and a fuzzy matching of reference data (source, volume, pagination, etc.). The matching percentage by journal is

given in *Figure 7—figure supplement 1*. All papers lacking full first-name information for one or more authors were excluded from the sample (*n*=362,453; 15.5% of the population sample). The name-to-gender assignment algorithm, Gender API (*Gender API, 2016*), was used to determine the gender of all authors per paper for the remaining sample. This algorithm estimates a given author's likelihood of being a man or a woman based on first name and country affiliation. The accuracy of the algorithm has previously been validated in a random subsample (N=500 authors) drawn from the same dataset (*Nielsen et al., 2017*), and was recently evaluated as the best-performing service, in a benchmark of five name-to-gender assignment algorithms (*Santamaría and Mihaljević, 2018*). Gender API provided valid name-to-gender estimates for 1,434,715 papers (61.4% of the population sample) (for further specification on Gender API, See *Figure 7—figure supplement 3*). A sensitivity analysis indicated unreliable Gender API estimates for authors from 18 countries, located in Eastern Asia and Sub-Saharan Africa. All documents with first and last authors

from these countries were excluded (see *Figure 7—figure supplement 2*). This reduced the sample by 7.2% (*n*=165,173), resulting in a final sample of 1,269,542 papers (54.3% of the population sample).

In the analysis of citation impact per paper, exact matching covariates were included for institutional prestige, geographical region and medical specialty. All three factors are known to influence citation impact (*Judge, 2016*; *Stremersch et al., 2007*; *van Eck et al., 2013*). In addition, research shows that the participation of women in medical research varies considerably across geographical regions, top and lower-tier research institutions and medical specialties (*Lautenberger et al., 2014*; *Nielsen et al., 2017*; *Weeden et al., 2017*). Matching of institutional prestige was based on a binary variable specifying whether a paper includes authors affiliated with a top-100 university according to the Leiden Ranking [www.leidenranking.com]. The matching of geographical region was based on ten variables specifying the location of the first and last author. The matching of medical specialties was based on 124 specialties identified using the HeTOP MeSH specialty-classification algorithm (*Darmoni et al., 2006*) (for specifications on country groupings and specialty-disambiguation, see *Figure 7—source datas 1–3*). We used replacement sampling, resulting in case and control groups of equal sizes.

Five covariates were included in the regression models. Journal prestige is arguably the strongest single predictor of a paper's citation impact (*Judge, 2016*). Prior work suggest that women are less likely than men to publish in journals with high impact factors (see, for example, *González-Álvarez and Cervera-Crespo, 2019*; *Lerchenmüller et al., 2018*). To adjust for this factor, we computed the mean NCS-score per journal (MNCS journal). This indicator is advantageous compared to the journal impact factor, most notably because it corrects for subfield-specific citation characteristics.

International collaboration is another recognized predictor of citation impact (*Smith et al., 2014*). Again, extant research suggests that women, on average, co-author fewer papers with international colleagues compared to men (see e.g. *Abramo et al., 2013*; *Larivière et al., 2013*). A binary variable adjusts for this factor (collaboration between authors from different countries = 1).

Finally, we included two count variables that adjust for the number of authors per paper and the number of per-paper self-citations within the

first four years after publication. Extant research demonstrates that per-paper citation impact is positively correlated with author-group size, and that women, on average, have fewer self-citations and fewer collaborators per paper (see e.g. *Araújo et al., 2017*; *King et al., 2017*).

A Tweedie distribution was used to estimate the relationship between author gender and NCS (*Funk et al., 2010*; *Jørgensen, 1987*). The continuous outcome variable, NCS, is highly right-skewed with a probability mass at zero. Tweedie distributions are a class of mixed compound Poisson-gamma distributions with a discrete mass at zero. This makes them useful for modeling continuous outcome variables with a mixture of zeros and positive values. Tweedie distributions belong to the exponential family of generalized linear models (GLM). The mean and variance for the Tweedie random variable are $E(Y)$ and $\mathrm{Var}(Y) = \varphi\mu^p$, respectively, where $\varphi$ is the dispersion parameter and $p$ is the parameter controlling the variance of the distribution. Tweedie distributions take variance-power values $p$ in the range >1 and < 2. We estimated three basic GLM-models using link power=0 corresponding to a log-link function and variance power of *p=1.65, p=1.72 and p=1.6* for the three models, F_first, F_last and F_both, respectively. Variance power was derived empirically through iterative algorithms seeking an optimal fit. The dispersion parameter was used to test for goodness of fit and examine possible overdispersion. Robustness checks were carried out to examine the sensitivity of the results to alternative model and sample specifications (see *Figure 2—source datas 2–4*). First, we ran negative binomial regressions with raw per-paper citation scores (with a four-year citation window) (CS) as the outcome variable in Samples 1, 2 and 3. Next, we ran Tweedie regressions with NCS as outcome variable based on the full, un-matched data set. Finally, we ran Tweedie regressions with dummy variables for different levels of MNCS journal (low, medium, and high). This allowed us to examine whether adjusting for journal prestige at different thresholds influenced the case coefficients in Samples 1, 2, and 3. The dummy variables were created based on percentile ranks of MNCS journal. The percentile thresholds were $\geq$ 95% for the high-category variable, $\geq$ 50% < 95% for the medium-category variable and < 50% for the low category variable.

Logistic regression was used to estimate the relationship between the four covariates (self-citations, N authors, MNCS journal and

international collaboration) and the case variable in each sample.

The predictors and covariates in all the regression models had Variance Inflation Factors below two, indicating very low levels of multicollinearity.

The statistical analyses were conducted in R version 3.4.3. For the matching procedure, we used the R "Matching" package (*Sekhon, 2011*), for the Tweedie regressions we used the "twee-die" and "statmod" packages (*Dunn, 2017*; *Dunn and Smyth, 2008*; *Dunn and Smyth, 2005*). Finally, we used the "emmeans' package to calculate the estimated marginal means for the case variables in the Tweedie regressions (*Lenth, 2019*).

Information for the calculation of bibliometric indices (CS, NCS, JS, MNCS journal, self-citations and international collaboration) were obtained from the Centre for Science and Technology Studies (CWTS), Leiden University. CWTS hosts a curated, quality-added version of the Web of Science which enables the calculation of field-normalized citation indicators, which is not immediately possible in the standard version available online. Calculation methods are standard operations, as described in *Waltman et al. (2012)*. For clarity, we briefly explain the NCS and MNCS journal indicators here. The purpose of using field-normalized citation indicators is to account for very large differences in citation activity and density across fields, stemming from differences in the referencing behavior and norms for various fields. The operation makes comparison between fields possible, as the score expresses impact relative to the field a given paper is published in, rather than an absolute impact, which may have different meanings across fields. To normalize citation scores, the raw citation count (CS) is divided by the mean citation scores of equivalent papers from the same field. These are papers published in the same year and field, and in this case, with citations counted in the same number of years. This gives us the NCS. Fields are here delimited by an algorithmic approach developed by *Waltman and van Eck (2012)*, where papers are assigned to clusters based on their citing, citation and topical commonalities. These clusters thus define small fields with common referencing cultures, increasing internal consistency when calculating field normalizations. The MNCS journal is simply the mean NCS of all papers published in a given journal in a given year. Like the Journal Impact Factor, the MNCS journal changes from one year to another, and the

MNCS journal for a paper is then calculated for the year the paper was published.

Weitzman's measure, or Δ, is well-defined for density functions. Let *f(x)* and *g(x)* be two probability density functions, then:

$$\Delta = \int \min(f(x), g(x))dx$$

However, for empirical distributions the solution is not as well-defined. We used the "overlapping" R-package (*Pastore, 2018*), which divides two empirical density distributions into intervals and calculates the cumulative sum (integral) of minimum values per interval. As both distributions by definition have a cumulative sum of 1, the result is in the range 0 to 1, where 1 implies identical distributions and 0 the complete absence of any overlap. Estimating the overlap empirically heavily depends on the number of bins the distribution is divided into. We tested various bin ranges for our samples and found estimates stabilized around 5,000 bins and upward, and thus used 10,000 bins for the analysis.

**Jens Peter Andersen** is in the Danish Centre for Studies on Research and Research Policy, Aarhus University, Aarhus, Denmark
jpa@ps.au.dk
https://orcid.org/0000-0003-2444-6210

**Jesper Wiborg Schneider** is in the Danish Centre for Studies on Research and Research Policy, Aarhus University, Aarhus, Denmark
https://orcid.org/0000-0001-5556-0919

**Reshma Jagsi** is in the Department of Radiation Oncology and the Center for Bioethics and Social Sciences in Medicine, University of Michigan, Ann Arbor, Michigan, United States
https://orcid.org/0000-0001-6562-1228

**Mathias Wullum Nielsen** is in the Danish Centre for Studies on Research and Research Policy, Aarhus University, Aarhus, Denmark
https://orcid.org/0000-0001-8759-7150

*Author contributions:* Jens Peter Andersen, Conceptualization, Resources, Data curation, Software, Formal analysis, Validation, Investigation, Visualization, Methodology, Writing—original draft, Project administration, Writing—review and editing; Jesper Wiborg Schneider, Conceptualization, Resources, Formal analysis, Funding acquisition, Validation, Investigation, Visualization, Methodology, Writing—original draft, Writing—review and editing; Reshma Jagsi, Conceptualization, Data curation, Formal analysis, Validation, Writing—original draft, Writing—review and editing; Mathias Wullum Nielsen, Conceptualization, Resources, Formal analysis, Funding acquisition, Validation,

Investigation, Visualization, Methodology, Writing—original draft, Project administration, Writing—review and editing

*Competing interests:* Reshma Jagsi: RJ: Stock options in Equity Quotient; advisory role and personal fees from Amgen; and consulting for Vizient. The other authors declare that no competing interests exist.

Funding

| Funder | Grant reference number | Author |
|---|---|---|
| Uddannelses- og Forskningsminis-teriet | 6183-00001B | Jesper Wiborg Schneider |
| Aarhus University Research Foundation | AUFF-2018-7-5 | Mathias Wullum Nielsen |

The funders had no role in study design, data collection and interpretation, or the decision to submit the work for publication.

Decision letter and Author response
Decision letter https://doi.org/10.7554/eLife.45374.026
Author response https://doi.org/10.7554/eLife.45374.027

# Additional files

## Supplementary files
• Supplementary file 1. Literature review.
DOI: https://doi.org/10.7554/eLife.45374.022
• Transparent reporting form
DOI: https://doi.org/10.7554/eLife.45374.023

## Data availability
All final data and analytical scripts are available on GitHub: https://github.com/ipoga/gendcit (copy archived at https://github.com/elifesciences-publications/gendcit).

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
