## [Decision Letter]

Thank you for submitting your article "No discernible gender difference found in global analysis of leading medical authors' per-paper citation impact" for consideration by eLife. Your article has been reviewed by four peer reviewers, and the evaluation has been overseen by the eLife Features Editor (Peter Rodgers). The following individuals involved in review of your submission have agreed to reveal their identity: Willem M Otte (Reviewer #1); Neven Caplar (Reviewer #2); Marc Lerchenmueller (Reviewer #3); Rinita Dam (Reviewer #4).

The reviewers have discussed the reviews with one another and the eLife Features Editor has drafted this decision to help you prepare a revised submission. We hope you will be able to submit the revised version within two months.

SUMMARY

The reviewers were positive about many aspects of the work. For instance, one wrote: "The authors are to be commended for an encompassing piece of work - literature review, data collection and analyses - on a very important and contentious topic in connection to the gender gap in science". However, the reviewers also raised a number of technical point that you will need to address in a revised version. You will also need to better describe the steps in your analysis and to revise the text in a number of places (see below).

LIST OF POINTS THAT NEED TO BE ADDRESSED

# General point

1. A central concern is that the manuscript in its current form convolutes confounding with mechanistic explanations for a potential sex difference in citations in the cross-section. To elaborate, prior work provides evidence that, for example, women relative to men collaborate less internationally and place papers in less prestigious journals. Both phenomena are associated with lower citations, with each then constituting a partial mechanistic explanation for why women may end up receiving less citations.

This differs in meaning from confounding a sex difference in citations that, once adjusting for, should not exist. In other words, if journal prestige adequately proxies the paper-level quality of the manuscripts that get accepted, and women's manuscripts would be accepted less frequently due to genuine quality differentials (versus e.g., bias in the review process), then the authors would have identified a partial explanation for sex differences in citations rather than having rectified a confounded result in prior literature.

Likewise if international collaborations produce higher quality science, as potentially indicated by higher citations, and women collaborate less and by extension produce different quality, on average, then one would expect lower citations to women's work. Overall, considering the matching and various "controls" employed across which men and women conceivably sort differently, one might wonder what residual effect of gender one would expect in the presented regressions?

# Introduction

2. Please provide references for the sentences that start:

- "As of 2013, women constituted 21% of full professors... . " (please add at least one reference)

- "Some studies report. . . " (please one or two references for each of the three classes of findings mentioned in this sentence)

- "Only five studies report. . . " (please add five references)

3. The measure used to quantify presence or absence of gender disparities in medical research, namely the citations, is a very specific proxy. My question would be whether this proxy captures the issue sufficiently. In countries like the Netherlands, for example, tenure track positions and grant scorings are not just based on number of citations or number of publications. The focus is often on the 'five key papers' of a person and on the impact this work has had on scientific and clinical progress. Sometimes a single paper has far more influence on a persons career, visibility and rewards than ten others. I understand that these things are currently not (yet) measurable on a large scale, but please discuss the limitations of using citation counts as a measure of the performance of scientists.

4. The paragraph that starts "However, in one of the most influential multidisciplinary studies on the topic.. . " is too long and should be shortened.

Please also delete the passage "However, others may interpret [... ] and research leaders (Casteevecchi, 2018; Sewell, 2018)." as it is too speculative for a peer-reviewed paper.

5. Please consider citing and briefly commenting upon following recent papers at appropriate places in the manuscript:

Sugimoto, Ahn, Smith, Macaluso and Larivière. 2019. Factors affecting sex-related reporting in medical research: a cross-disciplinary bibliometric analysis. The Lancet. DOI: https://doi.org/10.1016/S0140-6736(18)32995-7)

Gonzalez-Alvarez and Cervera-Crespo. 2019. Psychiatry research and gender diversity: authors, editors and peer reviewers. The Lancet Psychiatry DOI: https://doi.org/10.1016/S2215-0366(19)30039-2

6. The results of your study will be summarized in the abstract, so it is not necessary to summarize them again at the end of the introduction, so please delete the paragraph that starts "We find little evidence... "

# Results

7. Table 1 gives details about the number of papers with female authors etc in the five medical specialities considered in the survey; please consider including similar tables for geographic location and institutional prestige.

8. Please move the information about NCS from the methods section to the start of the paragraph that begins: "Figure 1 displays the.. . ".

This paragraph also needs to include more information about the distributions (eg, number of cases and controls for each sample, standard deviations for each distribution), and to avoid words like trivial and immense. Therefore, please replace the last six sentences in this paragraph with the following (or something similar).

"On average, papers with female first authors are cited X.X% less than papers with male first authors (sample 1. Female first authors: n = XXXX; x-bar = 1.16; s.d. = XXX; x-tilda =0.73. Male first authors: n = XXXX; x-bar = 1.27; s.d. = XXX; x-tilda =0.76. etc): however, the overlap between the two distributions is extensive (Cohen's d = -0.060; Weitzman's Delta = 95.4%; Weitzman, 1970). Papers with female first authors are cited X.X% less than papers with male last authors (sample 2. Female last authors: n = XXXX; x-bar = 1.16; s.d. = XXX; x-tilda =0.72. Male first authors: n = XXXX; x-bar = 1.24; s.d. = XXX; x-tilda =0.76. etc): again, the overlap between the two distributions is extensive (Cohen's d = -0.042; Weitzman's Delta = 95.6%). Papers in which both the first and last authors are female are cited 14.8% less than papers with other gender combinations (sample 3. Female first and last authors: n = XXXX; x-bar = 1.11; s.d. = XXX; x-tilda =0.71. Other combinations: n = XXXX; x-bar = 1.27; s.d. = XXX; x-tilda =0.76. etc): again, the overlap between the two distributions is extensive (Cohen's d = -0.081; Weitzman's Delta = 93.1%). While these results are consistent with previous reports of a citation advantage for papers by male authors, we decided to use regression analyses to explore if there were other explanations for the difference."

Also, for both distributions in each sample, please consider quoting the uncertainty on the mean (derived either under Gaussian approximation as standard deviation/sqrt(N-1), or via bootstrapping).

9. In the caption for figure 1, please explicitly state what the x- and y-axes are (including the units for the y-axis), and what the vertical dashed lines show.

10. In the paragraph that starts "Regression analysis were carried out.. . ", please explain how the field-normalized journal impact is calculated, and how the incidence ratio is calculated, and what incidence ratios of less than one and more than one mean.

11. The discussion of figure 3 in the paragraph that starts "To identify the primary factors.. . " is difficult to read: please move all the values for x-bar, x-tilda etc to the figure caption, and please use consistent terminology throughout (ie MNCS rather than journal impact).

12. A number of the reviewers found figure 3 confusing. Figure 2 suggests that MNCS is the largest contributor to the differences seen in figure 1, but figure 3 suggests that self-citation is the largest contributor. Please clarify.

13. Following on from this, it is not clear how the authors can claim that both self-citations and journal impact independently explain most of the average gender difference. This is possibly only in the case that 1. These two variables are tightly correlated or 2. When taking into account both of these parameters, men should have been receiving less citations than women? Just to expand on this point, I am especially confused with the self-citation statement, given the papers authored by men seems to have much larger tail of the papers with very high citations numbers (Figure 1) and I find it very difficult to believe that only self-citations could explain this high-citation tail of the distribution.

The authors may wish to consult and discuss the following study which, although spanning various fields and productivity levels, suggests self-citation rates to be about 5%-10% of overall citations (which seems similar to the study by King et al, 2017 that the authors cite):

Ioannidis JP, Klavans R, Boyack KW. 2016. Multiple citation indicators and their composite across scientific disciplines. PLOS Biology 14:e1002501.

14. One aspect that, to my mind, clouds the interpretation of the results is the way the authors account for "field". The matching accounts for field via 124 specialties and I understand from the description that the matching has been "exact" (not coarsened or nearest neighbour etc.) such that to be matched, authors had to be located in the same region, institution of like prestige, and same field. Now, the regressions account for field on both the left hand side and the right hand side of the equation through the WoS citation normalization (4K fields). It seems intuitive, that having the WoS normalization on both sides of the equation leads to substantial variance in NCS (your dependent variable) being explained by MNCS (independent variable), leaving little room (besides self-citations) for other covariates.

Related to this, it would be interesting to see the relative impact of MNCS in the regressions when the matching includes the 124 fields and when it does not. Likewise, it would be informative to control for journal through crude fixed effects (i.e., dummies for a specific journal) or journal impact factor tiers (e.g., low, middle, high JIF) in comparison to having the citation information embedded in your journal control variable to see how that influences the coefficient on gender in the regressions.

# Discussion

15. What lends credence to your interpretation of the results is, that you end up observing no tangible, residual effect of gender in your models with all controls despite the fact that the analysis is still in the cross-section and you likely pool men who are, on average, more experienced researchers than women. One might therefore expect men having a citation advantage in the cross-section. The authors might want to include this reflection in their discussion to bolster their interpretation of the findings.

# Methods

16. Based on personal experience with mapping of first and last author data [https://elifesciences.org/articles/34412] I know that the patterns of 'missing data' is not random. Most complete data is available for the most recent publications. The 'lag effect' - as is mentioned in the paper as well - may result in low citations scores for (senior) females just because the entered the field later than their (old) male colleagues. The authors try to mitigate the issue of time with a window approach, but there may still be an effect due to non-random missing data in the first years of the dataset. I would help if the authors could plot the percentages of missing data over time for the four gender classes.

17. The methods section needs to say more about how the following data were obtained and/or calculated:

- Institutional prestige

- NCS

- MNCS

- Self-citations

- International collaboration.

# Data and code

18. It would be very helpful, given the robustness of the analysis and general pipeline structure, if the authors would be willing to share their R code with the meta-research community on a (github) account.

19. Likewise, the authors should make available as much of their data as possible.

# Supplementary material

20. If your manuscript is accepted for publication we will explain how the information in the supplementary material file can be integrated into the paper itself, and how some items can be published as additional files.

[Editors' note: further revisions were requested prior to acceptance, as described below.]

Thank you for submitting the revised version of your article "No tangible gender difference found in global analysis of lead authors' per-paper citation impact in medical research" to eLife. The revised version of your article has been seen by three of the reviewers who reviewed the original version (Willem M Otte; Reviewer #1; Neven Caplar; Reviewer #2; Marc Lerchenmueller; Reviewer #3) and their responses have been largely positive. If you are able to address a small number final comments (see below), we will be able to accept your manuscript for publication.

REVIEWER COMMENTS:

Reviewer #1

I want to compliment the authors with this revised version of their manuscript. The work has significantly improved in terms of explanation, references to recent work by others as well as transparency (github) and deserves sharing with the community. I have no further issues or suggestions.

Reviewer #2

I believe that the authors have significantly improved their manuscript and I believe that I am now able to fully follow and understand how the analysis is conducted. I believe that the manuscript can be accepted in this form. I elaborate below.

I am still a bit surprised the authors choose to emphasize how the difference between the citation counts for men and women is "trivial" - if I were to write a same paper with the same data I would probably point out how the difference between citation counts for men and women, everything else being the same, is highly statistically significant. I would also be tempted to put actual numbers and error estimations in the abstract instead of quite abstract language that is currently there.

Having said that, I believe that authors have considerably improved their manuscript and the description of the analysis is now much clearer. My current objections are mostly stylistic. I see no obviously mistakes or problems in the analysis, and in the reported numbers - as such I am happy to recommend the manuscript for publication.

Note from Features Editor: Addressing the editorial comments below will address the concerns of Reviewer #2.

Reviewer #3

I would like to thank the authors for a thorough revision of the manuscript. I have no remaining substantive comment on the empirical work at this point. That said, please allow me to reiterate a concern I had when reading the initial submission of this work that remains in the revised version of the manuscript.

The authors report an unconditional mean sex difference in citations of 6.5% to 12.6% percent, which is in line with previous work. The authors then construct a set of control variables that allow testing for potential explanations for this gender gap and find, that especially self-citations and the impact of the publishing outlet have explanatory power. I appreciate that the authors have revised language in several locations that originally presented these effects as confounders to e.g., "co-variates". I also commend the authors on a thoughtful discussion of the self-citations (e.g., 4-year citation window, potentially more senior men than women in the cross-section). Still, the results allow for a narrative according to which women may garner fewer citations, at least in part, due to meritocratic dynamics. For example women may publish less frequently in high impact journals, which likely impedes the accumulation of citations, and there is emerging evidence on this dynamic (see Holman et al. 2018 or Lerchenmüller et al. 2018). Of course, it is conceivable that there exists e.g., gender bias in the review process, but until there is robust evidence for that conjecture (of which I am not aware of to date), it may also be that women experience lower citation counts because of a possibly meritocratic dynamic. Likewise, as the authors point out, sex differences in self-citations may stem from men being able to draw on more past work (particularly as the cross-section likely includes more men and the authors acknowledge that).

I would, therefore, encourage the authors to critically reflect about core claims that the manuscript currently makes, including "no tangible gender difference" in the title and "challenge meritocratic explanations" in the abstract. The fact is, that the data indicate an unconditional gender difference in the cross-section, apparently driven by the right hand tail (i.e., ostensible top performers). Although the difference is detected in the cross-section, even an ostensibly "small" difference of 6+% may accumulate to tangible career disadvantages over time (see Merton's work on the Matthew effect) and more longitudinal work is likely needed for more conclusive statements. The here identified mechanisms for this sex difference do not, to my mind, equate to there being no "tangible difference" but instead illuminate where a difference may stem from. Whereas it seems adequate to say that conditional on same quality journal, team size and international collaboration (perhaps as proxies for resources) etc. women produce work that is cited at a similar rate as work by men, it is not clear from the data that the unconditional effect does not exist for meritocratic dynamics. To be clear, I highlight these points with the belief that this manuscript has to offer a valuable contribution to a contentious and important topic, and I also believe that offering the most impartial interpretation may increase the community's recognition of this contribution.

- Holman, L., Stuart-Fox, D., & Hauser, C. E. (2018). The gender gap in science: How long until women are equally represented?. PLoS biology, 16(4), e2004956.

- Merton, R. K. (1968). The Matthew effect in science: The reward and communication systems of science are considered. Science, 159(3810), 56-63.

- Lerchenmüller, C., Lerchenmueller, M. J., & Sorenson, O. (2018). Long-term analysis of sex differences in prestigious authorships in cardiovascular research supported by the National Institutes of Health. Circulation, 137(8), 880-882.

1. Note from Features Editor: Addressing the editorial comments below will address most of the concerns of this Reviewer #3. However, please consider making further revisions to discuss and cite some or all of the papers by Holman et al., Merton, and Lerchenmüller et al.

EDITORIAL COMMENTS:

2. Please consider changing the title to the following, or something similar, to address the concerns of reviewers #2 and #3.

Gender difference in per-paper citation impact is mostly due to differences in self-citation and journal prestige

3. Please consider rewording the abstract as follows to address the concerns of reviewers #2 and #3:

A number of studies have found that scientific papers with women in leading-author positions attract fewer citations than those with men in leading-author positions. Here we report the results of a matched case-control study of 1,269,542 papers in selected areas of medicine published between 2008 and 2014. We find that papers with female authors are cited between 6.5% and 12.6% less than papers with male authors. However, when we adjust for self-citations, number of authors, international collaboration and journal prestige, we found near-identical per-paper citation impact for women and men in first and last author positions, with self-citations and journal prestige accounting for most of the difference. Given the underrepresentation of women in the upper echelons of academic medicine, these results highlight the importance of working to remove to the complex structural and cultural barriers that perpetuate gender inequalities in scientific organizations.

4. Please consider rewording the introduction as follows to address the concerns of reviewers #2 and #3 and to improve the flow of this section:

Over the past four decades, the share of female graduates in medicine has increased from less than 10% to more than 50% in OECD countries, and recent statistics suggest near-parity in the representation of women and men as authors in medical research in Australia, Brazil, Chile, Europe and North America (OECD, 2019; Elsevier, 2017). However, gender inequalities persist in the upper echelons of academic medicine. Moreover, as of 2013, women constituted just 21% of full professors in the United States and just 23% in Europe, with the proportion of women department chairs and deans being lower [OK?] (European Commission, 2016; Lautenberger et al., 2014).

These gender imbalances likely reflect myriad obstacles to women's career progress, including chilly and sometimes hostile work climates (Carr et al., 2003; Jenner et al., 2018; Pololi et al., 2013), bias in recruitment and selection practices (Van den Brink, 2011), societal cultures that still expect a strongly gendered division of domestic labor (Jolly et al., 2014), an underrepresentation of women in last-author positions (González-álvarez and Cervera-Crespo, 2019; Jagsi et al., 2006; Lerchenmueller and Sorenson, 2018), and disparities in research funding (Jagsi et al., 2009; Sege et al., 2015). Given that citation indicators are increasingly being used to inform tenure, hiring and funding decisions in many areas of the medical sciences, any gender bias in citations has the potential to contribute to the perpetuation of these inequalities, so a number of researchers have explored the topic of gender and citations in recent years.

A survey of the literature revealed 22 papers on gender and citations in the medical sciences published between 2006 and 2016 (see supplementary file 1). The study designs, impact measures and statistics used in these papers are too heterogeneous for meta-analytical comparisons, and this literature is also characterized by notable variations in results depending on specialty, country, study design and type of citation indicator (h-index, citations per paper, cumulative citations, m-quotient and journal impact factor). Some studies report an average male-citation advantage (e.g., Larivière et al., 2011; Nielsen, 2016), whereas others do not observe any notable gender difference (e.g., Mirnezami et al., 2016; Pagel and Hudetz, 2011). Existing articles are in most cases based on convenience samples and limit their focus to single specialties or sub-specialties (16 out of 22), and the literature is characterized by a North American, bias with only five studies focusing on countries outside the US and Canada. Moreover, most articles (14 out of 22) base their gender comparisons on relatively small samples, and very few adjust for relevant covariates that may contribute to explain average gender differences, such as collaboration patterns, numbers of authors per paper, self-citations and institutional prestige. Furthermore, only six of the papers report direct comparisons of the average number of citations per paper for male and female authors (Housri et al., 2008; Larivière et al., 2011; Mirnezami et al., 2016; Nielsen, 2016; Pagel and Hudetz, 2015; Pagel and Hudetz; 2011).

Researchers have also studied gender and bias in fields other than medicine, and again these studies are characterized by ambiguous results that vary by geographical focus, time-period and discipline. Some report differences in favor of male authors (Aksnes et al., 2011; Larivière et al., 2013; Caplar et al., 2017; Eagly and Miller, 2016; Maliniak et al., 2013), some report smaller differences in favour of female authors (Borrego et al., 2010; Long, 1992; van Arensbergen et al., 2012), and some report no discernable gender difference (Nielsen, 2017; Slyder et al., 2011; Symonds et al., 2006). [Query: Is it correct that Nielsen, 2016 finds a male-citation advantage, where Nielsen, 2017 finds no gender difference?]

Here we report the results of a comprehensive, global analysis of possible gender variations in the per-paper citation impact of medical researchers. We analyzed 1,269,542 papers on disease-specific medical research published between 2008 and 2014 (n=1,269,542). To reduce confounding and ensure balanced case-control groups, three matching covariates (institutional prestige, geographic location and medical specialty) were used to generate three datasets: sample 1 had female first authors as the case and male first authors as the control (n=1,018,665); sample 2 had female last authors as the case and male last authors as the control (n=653,233); and in sample 3, pairs of female first and last authors constituted the case group and all other author combinations were included in the control group (n=368,374). The outcome variable was field-normalized citations per paper, and regression analyzes were used to explore the influence of additional co-varying factors (self-citations, number of authors, international collaboration and journal prestige) on differences in per-paper citation impact (see Methods). Given the large sample size, global scope, and matched design, our study is less vulnerable to biases resulting from sample-specific variance, confounders and selection than previous studies.

5. The word "trivial" appears four times in your manuscript. Please reword (by, for example, changing trivial to small or a similar word) to address the concerns of reviewers #2 and #3. The first two sentences of the conclusion could also be reworded as follows:

In conclusion, our results demonstrate that, adjusting for co-varying factors, men and women in first and last author positions are cited at similar rates.

6. In general the manuscript refers to "per-paper citation impact" or just "citation impact". However, it sometimes uses other phrases, such as "citation score" or "citation rate". If these phrases all mean different things, that is fine. However, if any two of them mean the same thing, please use just one of them.

---

## [Author Response]

# General pointA central concern is that the manuscript in its current form convolutes confounding with mechanistic explanations for a potential sex difference in citations in the cross-section. To elaborate, prior work provides evidence that, for example, women relative to men collaborate less internationally and place papers in less prestigious journals. Both phenomena are associated with lower citations, with each then constituting a partial mechanistic explanation for why women may end up receiving less citations.This differs in meaning from confounding a sex difference in citations that, once adjusting for, should not exist. In other words, if journal prestige adequately proxies the paper-level quality of the manuscripts that get accepted, and women's manuscripts would be accepted less frequently due to genuine quality differentials (versus e.g., bias in the review process), then the authors would have identified a partial explanation for sex differences in citations rather than having rectified a confounded result in prior literature.Likewise if international collaborations produce higher quality science, as potentially indicated by higher citations, and women collaborate less and by extension produce different quality, on average, then one would expect lower citations to women's work. Overall, considering the matching and various "controls" employed across which men and women conceivably sort differently, one might wonder what residual effect of gender one would expect in the presented regressions?

These are excellent points. We have revised the paper to make this distinction clear and removed the term “confounding” where it is unwarranted.

# Introduction2. Please provide references for the sentences that start:- "As of 2013, women constituted 21% of full professors... . " (please add at least one reference)- "Some studies report. . . " (please one or two references for each of the three classes of findings mentioned in this sentence)- "Only five studies report. . . " (please add five references)

The requested references have been added in the revised manuscript.

3. The measure used to quantify presence or absence of gender disparities in medical research, namely the citations, is a very specific proxy. My question would be whether this proxy captures the issue sufficiently. In countries like the Netherlands, for example, tenure track positions and grant scorings are not just based on number of citations or number of publications. The focus is often on the 'five key papers' of a person and on the impact this work has had on scientific and clinical progress. Sometimes a single paper has far more influence on a persons career, visibility and rewards than ten others. I understand that these things are currently not (yet) measurable on a large scale, but please discuss the limitations of using citation counts as a measure of the performance of scientists.

This is a certainly a relevant perspective, and we have included a reflection on this matter in the Discussion section of the revised manuscript.

4. The paragraph that starts "However, in one of the most influential multidisciplinary studies on the topic.. . " is too long and should be shortened.Please also delete the passage "However, others may interpret [... ] and research leaders (Casteevecchi, 2018; Sewell, 2018)." as it is too speculative for a peer-reviewed paper.

The paragraph has been shortened, and we have removed the speculative passage in the introduction.

5. Please consider citing and briefly commenting upon following recent papers at appropriate places in the manuscript:Sugimoto, Ahn, Smith, Macaluso and Larivière. 2019. Factors affecting sex-related reporting in medical research: a cross-disciplinary bibliometric analysis. The Lancet. DOI: https://doi.org/10.1016/S0140-6736(18)32995-7)Gonzalez-Alvarez and Cervera-Crespo. 2019. Psychiatry research and gender diversity: authors, editors and peer reviewers. The Lancet PsychiatryDOI: https://doi.org/10.1016/S2215-0366(19)30039-2

In the revised manuscript, we cite and briefly comment upon each of these references.

6. The results of your study will be summarized in the abstract, so it is not necessary to summarize them again at the end of the introduction, so please delete the paragraph that starts "We find little evidence... "

Good point. We have removed this section from the introduction.

# Results7. Table 1 gives details about the number of papers with female authors etc in the five medical specialities considered in the survey; please consider including similar tables for geographic location and institutional prestige.

Thank you for this suggestion. Table 1 now includes information about geographic location and institutional prestige as well.

8. Please move the information about NCS from the methods section to the start of the paragraph that begins: "Figure 1 displays the.. . ".This paragraph also needs to include more information about the distributions (eg, number of cases and controls for each sample, standard deviations for each distribution), and to avoid words like trivial and immense. Therefore, please replace the last six sentences in this paragraph with the following (or something similar)."On average, papers with female first authors are cited X.X% less than papers with male first authors (sample 1. Female first authors: n = XXXX; x-bar = 1.16; s.d. = XXX; x-tilda =0.73. Male first authors: n = XXXX; x-bar = 1.27; s.d. = XXX; x-tilda =0.76. etc): however, the overlap between the two distributions is extensive (Cohen's d = -0.060; Weitzman's Delta = 95.4%; Weitzman, 1970). Papers with female first authors are cited X.X% less than papers with male last authors (sample 2. Female last authors: n = XXXX; x-bar = 1.16; s.d. = XXX; x-tilda =0.72. Male first authors: n = XXXX; x-bar = 1.24; s.d. = XXX; x-tilda =0.76. etc): again, the overlap between the two distributions is extensive (Cohen's d = -0.042; Weitzman's Delta = 95.6%). Papers in which both the first and last authors are female are cited 14.8% less than papers with other gender combinations (sample 3. Female first and last authors: n = XXXX; x-bar = 1.11; s.d. = XXX; x-tilda =0.71. Other combinations: n = XXXX; x-bar = 1.27; s.d. = XXX; x-tilda =0.76. etc): again, the overlap between the two distributions is extensive (Cohen's d = -0.081; Weitzman's Delta = 93.1%). While these results are consistent with previous reports of a citation advantage for papers by male authors, we decided to use regression analyses to explore if there were other explanations for the difference."Also, for both distributions in each sample, please consider quoting the uncertainty on the mean (derived either under Gaussian approximation as standard deviation/sqrt(N-1), or via bootstrapping).

These are great points. The manuscript has been revised in accordance with each of the requests. For all distributions, the absolute uncertainty of the mean is between 0.001 and 0.005. This is reported in the first paragraph, p. 4.

9. In the caption for figure 1, please explicitly state what the x- and y-axes are (including the units for the y-axis), and what the vertical dashed lines show.

The caption for Figure 1 has been revised in accordance with these requests.

10. In the paragraph that starts "Regression analysis were carried out.. . ", please explain how the field-normalized journal impact is calculated, and how the incidence ratio is calculated, and what incidence ratios of less than one and more than one mean.

These are great suggestions. This paragraph now includes a specification on how NCS and MNCS journal are calculated. We also specify how the exponentiated coefficients in the Tweedie should be interpreted. In addition, we have computed estimated marginal means to allow for a more intuitive interpretation of the outcomes of the Tweedie regressions. Please note that that the numeric regression inputs have been rescaled by dividing by two standard deviations to allow for meaningful comparison of binary and numeric variables. We have also expanded the field-normalization explanation in the methods section.

11. The discussion of figure 3 in the paragraph that starts "To identify the primary factors.. . " is difficult to read: please move all the values for x-bar, x-tilda etc to the figure caption, and please use consistent terminology throughout (ie MNCS rather than journal impact).

This is a good point. This information is now reported in Table 2. Please note that we have revised this part of the manuscript considerably to accommodate other reviewer concerns.

12. A number of the reviewers found figure 3 confusing. Figure 2 suggests that MNCS is the largest contributor to the differences seen in figure 1, but figure 3 suggests that self-citation is the largest contributor. Please clarify.13. Following on from this, it is not clear how the authors can claim that both self-citations and journal impact independently explain most of the average gender difference. This is possibly only in the case that 1. These two variables are tightly correlated or 2. When taking into account both of these parameters, men should have been receiving less citations than women? Just to expand on this point, I am especially confused with the self-citation statement, given the papers authored by men seems to have much larger tail of the papers with very high citations numbers (Figure 1) and I find it very difficult to believe that only self-citations could explain this high-citation tail of the distribution.The authors may wish to consult and discuss the following study which, although spanning various fields and productivity levels, suggests self-citation rates to be about 5%-10% of overall citations (which seems similar to the study by King et al, 2017 that the authors cite):Ioannidis JP, Klavans R, Boyack KW. 2016. Multiple citation indicators and their composite across scientific disciplines. PLOS Biology 14:e1002501.

These are very important points. We have removed Figure 3 from the manuscript. In the revised version, we use logistic regression to examine associations between the covariates and the case variables (Figure 4). In figure 5, we examine the average proportion of per-paper self-citations plotted by NCS-based quantiles (5-percentile bins) in samples 1, 2, 3. This figure indicates that self-citations comprise approximately 15% of the per-paper citations in the top 5% most cited papers. This implies that at least part of the gender variation observed on the right side of the curves in Figure 1 may be attributable to average gender differences in per-paper self-citation rates. Note here, that our citation indicators are calculated with a four-year window, which may contribute to explain the relatively large proportion of self-citations in the samples.

The revised manuscript also includes a more detailed analysis of variations in the representation of case papers and control papers plotted by MNCS journal-based 5-percentile bins – from low impact to high impact journals. Figure 6 indicates an underrepresentation of case papers (with women in leading author positions) in the top-5 percent journals with the highest MNCS-journal score.

14. One aspect that, to my mind, clouds the interpretation of the results is the way the authors account for "field". The matching accounts for field via 124 specialties and I understand from the description that the matching has been "exact" (not coarsened or nearest neighbour etc.) such that to be matched, authors had to be located in the same region, institution of like prestige, and same field. Now, the regressions account for field on both the left hand side and the right hand side of the equation through the WoS citation normalization (4K fields). It seems intuitive, that having the WoS normalization on both sides of the equation leads to substantial variance in NCS (your dependent variable) being explained by MNCS (independent variable), leaving little room (besides self-citations) for other covariates.Related to this, it would be interesting to see the relative impact of MNCS in the regressions when the matching includes the 124 fields and when it does not. Likewise, it would be informative to control for journal through crude fixed effects (i.e., dummies for a specific journal) or journal impact factor tiers (e.g., low, middle, high JIF) in comparison to having the citation information embedded in your journal control variable to see how that influences the coefficient on gender in the regressions.

In the original manuscript, we also ran the Tweedie regressions with raw, un-normalized citations scores as the outcome variables (as a robustness check) with qualitatively similar results. This suggests that double-normalization is not a problematic issue in this case. In the revised manuscript, we have included three robustness checks to examine the sensitivity of the results to alternative model and sample specifications. First, we ran negative binomial regressions with raw per-paper citation scores (with a four-year citation window) (CS) as the outcome variable in Samples 1, 2 and 3. Next, we ran Tweedie regressions with NCS as outcome variable based on the full, un-matched data set. Finally, we ran Tweedie regressions with dummy variables for different levels of journal prestige as suggested by one of the reviewers.

We should perhaps also stress that fields in field-normalizations and medical specialties are two very different things. We have tried to stick to the precise use of these terms and have added additional explanation about how field-normalizations are performed.

# Discussion15. What lends credence to your interpretation of the results is, that you end up observing no tangible, residual effect of gender in your models with all controls despite the fact that the analysis is still in the cross-section and you likely pool men who are, on average, more experienced researchers than women. One might therefore expect men having a citation advantage in the cross-section. The authors might want to include this reflection in their discussion to bolster their interpretation of the findings.

We agree that this is an important factor. We now reflect on this in the Discussion section.

# Methods16. Based on personal experience with mapping of first and last author data [https://elifesciences.org/articles/34412] I know that the patterns of 'missing data' is not random. Most complete data is available for the most recent publications. The 'lag effect' - as is mentioned in the paper as well - may result in low citations scores for (senior) females just because the entered the field later than their (old) male colleagues. The authors try to mitigate the issue of time with a window approach, but there may still be an effect due to non-random missing data in the first years of the dataset. I would help if the authors could plot the percentages of missing data over time for the four gender classes.

This is a relevant concern, which we have looked into. The proportion of papers for which we can reliably determine the gender is stable throughout the period, regardless of whether we compare it to the population (pubmed) or the matched WoS sample. We have added a figure showing this in the supplementary materials. There is however a slightly smaller proportion in 2008, the earliest year, suggesting this would be worse going further back in time.

17. The methods section needs to say more about how the following data were obtained and/or calculated:- Institutional prestige- NCS- MNCS- Self-citations- International collaboration.

We have added additional explanations of the calculation and acquisition of these variables, where they are used and more extensively in the methods section.

# Data and code18. It would be very helpful, given the robustness of the analysis and general pipeline structure, if the authors would be willing to share their R code with the meta-research community on a (github) account.19. Likewise, the authors should make available as much of their data as possible.

We completely agree in this practice and had planned to do something along this line. We have removed elements of our data (Web of Science identification numbers) that would prevent us from sharing the data, and have added all analytical files and cleaned, prepared datasets to a GitHub account. We are contractually prohibited from sharing the scripts that generate these datasets and the underlying raw data, as these are the property of Clarivate Analytics.

Data and code are available here: https://github.com/ipoga/gendcit

Compiled supplementary material notebook is available here: https://ipoga.github.io/gendcit/

# Supplementary material20. If your manuscript is accepted for publication we will explain how the information in the supplementary material file can be integrated into the paper itself, and how some items can be published as additional files.

As part of the above code sharing, we have also created a GitHub page containing a generated report with all the information from the supplementary materials. We hope this approach is useful and acceptable.

[Editors' note: further revisions were requested prior to acceptance, as described below.]

We have very carefully considered all the suggestions and chosen to adopt almost all of them. In the following, we respond directly to the comments with the changes we have made. During this process we have also made minor aesthetic revisions of the figures (purely to increase readability) and language – but no data or meaning has been changed apart from what has been suggested or as direct consequence of suggested changes.

Reviewer #1I want to compliment the authors with this revised version of their manuscript. The work has significantly improved in terms of explanation, references to recent work by others as well as transparency (github) and deserves sharing with the community. I have no further issues or suggestions.

Thank you for your kind review, and for previous work on the manuscript. We greatly appreciate it.

Reviewer #2I believe that the authors have significantly improved their manuscript and I believe that I am now able to fully follow and understand how the analysis is conducted. I believe that the manuscript can be accepted in this form. I elaborate below.I am still a bit surprised the authors choose to emphasize how the difference between the citation counts for men and women is ``trivial' - if I were to write a same paper with the same data I would probably point out how the difference between citation counts for men and women, everything else being the same, is highly statistically significant. I would also be tempted to put actual numbers and error estimations in the abstract instead of quite abstract language that is currently there.Having said that, I believe that authors have considerably improved their manuscript and the description of the analysis is now much clearer. My current objections are mostly stylistic. I see no obviously mistakes or problems in the analysis, and in the reported numbers - as such I am happy to recommend the manuscript for publication.Note from Features Editor: Addressing the editorial comments below will address the concerns of Reviewer #2.

Thank you for your comments, we are happy that the revisions have improved the manuscript. We have reconsidered the use of the word trivial in regard to effect sizes, and while it is essentially stylistic, we have changed it to “very small”. Originally, we aligned our interpretation with the rules of thumb outlined by Jacob Cohen where a standardized mean difference of (Cohen’s d) <.2 (corresponding to an odds ratio below 1.4) would be seen as trivial in size, not big enough to register as a small effect. However, as these rules of thumb are indeed contextual, and the term “trivial” may have problematic connotations in our context, we have decided to change it to “very small”.

Regarding the second suggestion (about statistical significance), in line with the American Statistical Association and others (see Wasserstein, Schirm & Lazar, 2019). We generally do not approve of the term “statistical significance” and its implications. We endorse an estimation approach. In the present study, we provide confidence intervals, although they have their limitations too. Nonetheless, with the sample sizes we work with, estimation becomes very precise and therefore the confidence limits become tiny.

Wasserstein, R. L., Schirm, A. L., & Lazar, N. A. (2019). Moving to a world beyond ‘p<0.05’. The American Statistician, 73(supol 1), 1-19

Reviewer #3I would like to thank the authors for a thorough revision of the manuscript. I have no remaining substantive comment on the empirical work at this point. That said, please allow me to reiterate a concern I had when reading the initial submission of this work that remains in the revised version of the manuscript.The authors report an unconditional mean sex difference in citations of 6.5% to 12.6% percent, which is in line with previous work. The authors then construct a set of control variables that allow testing for potential explanations for this gender gap and find, that especially self-citations and the impact of the publishing outlet have explanatory power. I appreciate that the authors have revised language in several locations that originally presented these effects as confounders to e.g., "co-variates". I also commend the authors on a thoughtful discussion of the self-citations (e.g., 4-year citation window, potentially more senior men than women in the cross-section). Still, the results allow for a narrative according to which women may garner fewer citations, at least in part, due to meritocratic dynamics. For example women may publish less frequently in high impact journals, which likely impedes the accumulation of citations, and there is emerging evidence on this dynamic (see Holman et al. 2018 or Lerchenmüller et al. 2018). Of course, it is conceivable that there exists e.g., gender bias in the review process, but until there is robust evidence for that conjecture (of which I am not aware of to date), it may also be that women experience lower citation counts because of a possibly meritocratic dynamic. Likewise, as the authors point out, sex differences in self-citations may stem from men being able to draw on more past work (particularly as the cross-section likely includes more men and the authors acknowledge that).I would, therefore, encourage the authors to critically reflect about core claims that the manuscript currently makes, including "no tangible gender difference" in the title and "challenge meritocratic explanations" in the abstract. The fact is, that the data indicate an unconditional gender difference in the cross-section, apparently driven by the right hand tail (i.e., ostensible top performers). Although the difference is detected in the cross-section, even an ostensibly "small" difference of 6+% may accumulate to tangible career disadvantages over time (see Merton's work on the Matthew effect) and more longitudinal work is likely needed for more conclusive statements. The here identified mechanisms for this sex difference do not, to my mind, equate to there being no "tangible difference" but instead illuminate where a difference may stem from. Whereas it seems adequate to say that conditional on same quality journal, team size and international collaboration (perhaps as proxies for resources) etc. women produce work that is cited at a similar rate as work by men, it is not clear from the data that the unconditional effect does not exist for meritocratic dynamics. To be clear, I highlight these points with the belief that this manuscript has to offer a valuable contribution to a contentious and important topic, and I also believe that offering the most impartial interpretation may increase the community's recognition of this contribution.- Holman, L., Stuart-Fox, D., & Hauser, C. E. (2018). The gender gap in science: How long until women are equally represented?. PLoS biology, 16(4), e2004956.- Merton, R. K. (1968). The Matthew effect in science: The reward and communication systems of science are considered. Science, 159(3810), 56-63.- Lerchenmüller, C., Lerchenmueller, M. J., & Sorenson, O. (2018). Long-term analysis of sex differences in prestigious authorships in cardiovascular research supported by the National Institutes of Health. Circulation, 137(8), 880-882.

Thank you for your insightful and very important comment. We want to emphasize that we acknowledge the unconditional gender differences in mean citation impact, and the potential harm this can cause for the individual researcher. However, noise in the underlying data (e.g. from unmatched citations) also means that there is little practical significance of the unconditional differences, when they are not larger than observed, although the sample size deceptively leads to statistical significance.

We also think it is valuable to point out that the observed, unconditional differences are not due to women deliberately being cited less (regardless of interpretation), but rather that it is explained by other factors, such as research area and outlet. Our emphasis is therefore also on the distributions and their overlap rather than the differences in means – meaning there are many more men and women who are the same than who are not.

We especially appreciated your final comment, which has led us to reconsider a number of phrasings, among which is a rephrasing of “trivial” to “very small”, for effects sizes.

1. Note from Features Editor: Addressing the editorial comments below will address most of the concerns of this Reviewer #3. However, please consider making further revisions to discuss and cite some or all of the papers by Holman et al., Merton, and Lerchenmüller et al.

We have included Lerchenmüller et al. We have not included Merton, as we believe our reference to Cole & Singer is more on point. We were not able to find a good location to include Holman et al. Other revisions are commented below.

EDITORIAL COMMENTS:

2. Please consider changing the title to the following, or something similar, to address the concerns of reviewers #2 and #3.Gender difference in per-paper citation impact is mostly due to differences in self-citation and journal prestige

We have revised the title in line with, but not identical to, the suggested title.

3. Please consider rewording the abstract as follows to address the concerns of reviewers #2 and #3:A number of studies have found that scientific papers with women in leading-author positions attract fewer citations than those with men in leading-author positions. Here we report the results of a matched case-control study of 1,269,542 papers in selected areas of medicine published between 2008 and 2014. We find that papers with female authors are cited between 6.5% and 12.6% less than papers with male authors. However, when we adjust for self-citations, number of authors, international collaboration and journal prestige, we found near-identical per-paper citation impact for women and men in first and last author positions, with self-citations and journal prestige accounting for most of the difference. Given the underrepresentation of women in the upper echelons of academic medicine, these results highlight the importance of working to remove to the complex structural and cultural barriers that perpetuate gender inequalities in scientific organizations.

We have revised the abstract to this version, albeit with minor modifications.

4. Please consider rewording the introduction as follows to address the concerns of reviewers #2 and #3 and to improve the flow of this section:Over the past four decades, the share of female graduates in medicine has increased from less than 10% to more than 50% in OECD countries, and recent statistics suggest near-parity in the representation of women and men as authors in medical research in Australia, Brazil, Chile, Europe and North America (OECD, 2019; Elsevier, 2017). However, gender inequalities persist in the upper echelons of academic medicine. Moreover, as of 2013, women constituted just 21% of full professors in the United States and just 23% in Europe, with the proportion of women department chairs and deans being lower [OK?] (European Commission, 2016; Lautenberger et al., 2014).These gender imbalances likely reflect myriad obstacles to women's career progress, including chilly and sometimes hostile work climates (Carr et al., 2003; Jenner et al., 2018; Pololi et al., 2013), bias in recruitment and selection practices (Van den Brink, 2011), societal cultures that still expect a strongly gendered division of domestic labor (Jolly et al., 2014), an underrepresentation of women in last-author positions (González-álvarez and Cervera-Crespo, 2019; Jagsi et al., 2006; Lerchenmueller and Sorenson, 2018), and disparities in research funding (Jagsi et al., 2009; Sege et al., 2015). Given that citation indicators are increasingly being used to inform tenure, hiring and funding decisions in many areas of the medical sciences, any gender bias in citations has the potential to contribute to the perpetuation of these inequalities, so a number of researchers have explored the topic of gender and citations in recent years.A survey of the literature revealed 22 papers on gender and citations in the medical sciences published between 2006 and 2016 (see supplementary file 1). The study designs, impact measures and statistics used in these papers are too heterogeneous for meta-analytical comparisons, and this literature is also characterized by notable variations in results depending on specialty, country, study design and type of citation indicator (h-index, citations per paper, cumulative citations, m-quotient and journal impact factor). Some studies report an average male-citation advantage (e.g., Larivière et al., 2011; Nielsen, 2016), whereas others do not observe any notable gender difference (e.g., Mirnezami et al., 2016; Pagel and Hudetz, 2011). Existing articles are in most cases based on convenience samples and limit their focus to single specialties or sub-specialties (16 out of 22), and the literature is characterized by a North American, bias with only five studies focusing on countries outside the US and Canada. Moreover, most articles (14 out of 22) base their gender comparisons on relatively small samples, and very few adjust for relevant covariates that may contribute to explain average gender differences, such as collaboration patterns, numbers of authors per paper, self-citations and institutional prestige. Furthermore, only six of the papers report direct comparisons of the average number of citations per paper for male and female authors (Housri et al., 2008; Larivière et al., 2011; Mirnezami et al., 2016; Nielsen, 2016; Pagel and Hudetz, 2015; Pagel and Hudetz; 2011).Researchers have also studied gender and bias in fields other than medicine, and again these studies are characterized by ambiguous results that vary by geographical focus, time-period and discipline. Some report differences in favor of male authors (Aksnes et al., 2011; Larivière et al., 2013; Caplar et al., 2017; Eagly and Miller, 2016; Maliniak et al., 2013), some report smaller differences in favour of female authors (Borrego et al., 2010; Long, 1992; van Arensbergen et al., 2012), and some report no discernable gender difference (Nielsen, 2017; Slyder et al., 2011; Symonds et al., 2006). [Query: Is it correct that Nielsen, 2016 finds a male-citation advantage, where Nielsen, 2017 finds no gender difference?]Here we report the results of a comprehensive, global analysis of possible gender variations in the per-paper citation impact of medical researchers. We analyzed 1,269,542 papers on disease-specific medical research published between 2008 and 2014 (n=1,269,542). To reduce confounding and ensure balanced case-control groups, three matching covariates (institutional prestige, geographic location and medical specialty) were used to generate three datasets: sample 1 had female first authors as the case and male first authors as the control (n=1,018,665); sample 2 had female last authors as the case and male last authors as the control (n=653,233); and in sample 3, pairs of female first and last authors constituted the case group and all other author combinations were included in the control group (n=368,374). The outcome variable was field-normalized citations per paper, and regression analyzes were used to explore the influence of additional co-varying factors (self-citations, number of authors, international collaboration and journal prestige) on differences in per-paper citation impact (see Methods). Given the large sample size, global scope, and matched design, our study is less vulnerable to biases resulting from sample-specific variance, confounders and selection than previous studies.

We are very grateful for the thorough help provided here. We have included the majority of the suggested changes and agree that the flow has improved as a result. The proposed revisions involved deleting a paragraph, which could be considered a more confrontational paragraph than the rest. Rather than deleting this paragraph we have rewritten it to be less confrontational. See page 2 line 35 with simple markup or mid-page 3 with full markup.

5. The word "trivial" appears four times in your manuscript. Please reword (by, for example, changing trivial to small or a similar word) to address the concerns of reviewers #2 and #3. The first two sentences of the conclusion could also be reworded as follows:In conclusion, our results demonstrate that, adjusting for co-varying factors, men and women in first and last author positions are cited at similar rates.

We have changed “trivial” and some occurrences of “marginal” to the probably more fitting “very small”. We have also reworded the line in question.

6. In general the manuscript refers to "per-paper citation impact" or just "citation impact". However, it sometimes uses other phrases, such as "citation score" or "citation rate". If these phrases all mean different things, that is fine. However, if any two of them mean the same thing, please use just one of them.

There are distinct differences in the interpretation of these terms. Impact is the general term, while score is used for specific indicator calculations. There were cases where “rates” was used in a less optimal fashion – these have now been corrected, and the remaining occurrences are correct.